# Sialidases and fucosidases of *Akkermansia muciniphila* are crucial for growth on mucin and nutrient sharing with mucus-associated gut bacteria

Bashar Shuoker[1,2,7], Michael J. Pichler [1,7], Chunsheng Jin [3], Hiroka Sakanaka[1], Haiyang Wu[4], Ana Martínez Gascueña [4], Jining Liu[5], Tine Sofie Nielsen[1], Jan Holgersson[5], Eva Nordberg Karlsson [2], Nathalie Juge [4], Sebastian Meier [6], Jens Preben Morth [1] ✉, Niclas G. Karlsson [3] & Maher Abou Hachem [1] ✉

The mucolytic human gut microbiota specialist *Akkermansia muciniphila* is proposed to boost mucin-secretion by the host, thereby being a key player in mucus turnover. Mucin glycan utilization requires the removal of protective caps, notably fucose and sialic acid, but the enzymatic details of this process remain largely unknown. Here, we describe the specificities of ten *A. muciniphila* glycoside hydrolases, which collectively remove all known sialyl and fucosyl mucin caps including those on double-sulfated epitopes. Structural analyses revealed an unprecedented fucosidase modular arrangement and explained the sialyl T-antigen specificity of a sialidase of a previously unknown family. Cell-attached sialidases and fucosidases displayed mucin-binding and their inhibition abolished growth of *A. muciniphila* on mucin. Remarkably, neither the sialic acid nor fucose contributed to *A. muciniphila* growth, but instead promoted butyrate production by co-cultured Clostridia. This study brings unprecedented mechanistic insight into the initiation of mucin *O*-glycan degradation by *A. muciniphila* and nutrient sharing between mucus-associated bacteria.

The human gut microbiota exerts a major impact on our immune and metabolic homeostasis[1,2]. The host's first line of defense against microbial insult is the intestinal mucosal barrier that has increased thickness towards the colon[3,4]. Mucins are the main structural and gel-forming scaffolds of the mucosa, which is dominated by Mucin 2 (MUC2) in the colon[5]. Similarly to other mucins, MUC2, is an *O*-glycoprotein that is secreted by intestinal goblet cells and consists of up to 80% (w/w) glycan chains[6] that exhibit large structural diversity (>100 structures reported)[7]. The *O*-glycan epitopes in mucin exhibit longitudinal variations along the gastrointestinal tract (GIT)[8]. The outer mucus surface offers a steady nutritional resource and adhesion sites for adapted microbiota groups, while the inner mucus layer in the colon is sterile. In humans, *O*-glycans in the small intestine and cecum regions are densely fucosylated, which decreases gradually toward the

[1]Department of Biotechnology and Biomedicine, Technical University of Denmark, Lyngby 2800, Denmark. [2]Biotechnology, Department of Chemistry, Lund University, Lund, Sweden. [3]Proteomics Core Facility at Sahlgrenska Academy, University of Gothenburg, Gothenburg, Sweden. [4]Quadram Institute Bioscience, Norwich, UK. [5]Department of Laboratory Medicine, Institute of Biomedicine, Sahlgrenska Academy, University of Gothenburg, Gothenburg, Sweden. [6]Department of Chemistry, Technical University of Denmark, Kgs Lyngby, Denmark. [7]These authors contributed equally: Bashar Shuoker, Michael J. Pichler. ✉e-mail: premo@dtu.dk; maha@bio.dtu.dk

distal colon, whereas an increasing gradient of sialylation and sulphation[7,8] is observed. The reverse fucosylation/sialylation gradients are observed in mice[9]. Recently, a single extracellular sulphatase was shown to be critical for the growth of *Bacteroides* spp. on densely-sulfated mucin[10]. Similarly, the presence of a specific sialidase is critical to growth of *Ruminococcus gnavus* on mucin[11].

Only a few gut microbiota members can grow on mucin as a sole carbon source[3,12]. Atypically, *Akkermansia muciniphila* relies solely on mucin and related host-derived glycans for growth[13], which is reflected by its large carbohydrate-active enzyme (CAZyme) repertoire[14]. *A. muciniphila* has received extensive attention due to its relative abundance in healthy hosts, as opposed to patients of gut inflammatory bowel disease, including Crohn's disease and ulcerative colitis (UC)[15], and obesity[16]. A positive association of *A. muciniphila* with Parkinson disease was also reported[17], suggesting complex interactions with the mucus-associated microbiota, manifested beyond the gut niche[18]. Several *A. muciniphila* CAZymes have been structurally and biochemically characterized using simple oligosaccharide substrates[19,20]. By contrast, the specificities of *A. muciniphila* exoglycosidases, and notably the sialic acid and fucose decapping apparatus, towards mucin *O*-glycans remain unexplored.

Here, we used a panel of mucins to characterize the enzymes that collectively grant *A. muciniphila* access to all known fucosyl- and sialyl-mucin epitopes. Biochemical, structural, and microbiological studies allowed us to identify the key mucin-decapping fucosidases and sialidases. We also investigated the contribution of the characterized enzyme panel to growth on mucin and on sharing the released monosaccharide caps with mucus-associated Bacillota (previously Firmicutes). Our findings promote a mechanistic understanding of the initial steps of mucin turnover by *A. muciniphila* and the importance of this process in supporting other members of the mucus-adherent microbiota.

## Results

### *A. muciniphila* encodes six divergent fucosidases

The genome of *A. muciniphila* encodes six fucosidases, four assigned into glycoside hydrolase family 29 (GH29, harbours α1,2/3/4/6-fucosidases) and two into GH95 (harbours α1,2-fucosidases) in the CAZy database[21]. These enzymes are henceforth designated as *Am*GH29A (locus tag Amuc_0010), *Am*GH29B (Amuc_0146) *Am*GH29C (Amuc_0392), *Am*GH29D (Amuc_0846), *Am*GH95A (Amuc_0186) and *Am*GH95B (Amuc_1120) (Supplementary Fig. 1a). All enzymes possess signal peptides, indicating non-cytoplasmic localisation (Supplementary Table 1). The GH29 enzymes have variable architectures, with *Am*GH29C and *Am*GH29D being the most complex and possessing two putative carbohydrate binding modules (CBMs) (Supplementary Fig. 1a, Supplementary Fig. 2). The fucosidase catalytic modules exhibit high sequence diversity (Supplementary Fig 1b) and populate hitherto undescribed clusters in the phylogenetic trees of GH29 and GH95 sequences in CAZy (Supplementary Fig. 3).

### Two key enzymes responsible for the defucosylation of mucin and structurally related glycans

We produced and purified all six full-length enzymes and determined their kinetic parameters towards *para*-nitrophenyl-α-ʟ-fucoside (*p*NPFuc) (Supplementary Table 2). Next, we assayed the enzymes against a panel of fuco-oligosaccharides. The main activity of *Am*GH29A and *Am*GH29B was on the Fucα1,3GlcNAc disaccharide (Supplementary Fig. 4a). By contrast, *Am*GH29C and *Am*GH29D were active on larger human milk oligosaccharide (HMOs) and Lewis (Le) epitopes (Supplementary Fig. 4a–d, f–j). Both *Am*GH29C and *Am*GH29D hydrolysed 3-fucosyl lactose (3FL), but only *Am*GH29C could access this motif in the extended structure LNFP V (Supplementary Fig. 4f, k, m). However, the profiles of these two enzymes towards α-1,4-fucosyl in Le[a/b] motifs were similar (Supplementary

Fig. 4j–l). Key differences between *Am*GH29C and *Am*GH29D were the activity of *Am*GH29C but not *Am*GH29D towards the sialyl Le[a] tetrasaccharide (Supplementary Fig. 4n) and the low activity of *Am*GH29D towards 2´FL (Supplementary Fig. 4e).

The two GH95 enzymes were active on Fucα1,2Gal and 2´FL (Supplementary Fig. 4c, e). Only *Am*GH95A exhibited activity on Fucα1,3GlcNAc, but not on the galactosyl-extended Le[x] epitope (Supplementary Fig. 4a, h). By contrast, only *Am*GH95B hydrolyzed α1,2Fuc linkages in the Le[b] tetra- and hexasaccharides (Supplementary Fig. 4i, j) and additionally hydrolysed α1,3Fuc linkages in 3FL (Supplementary Fig. 4f). These data indicate that *Am*GH95B possesses a broader substrate range than *Am*GH95A.

Hitherto reported regioselectivities of enzymes in GH29[22] and GH95[23,24] stem from measurements on simple model oligosaccharides. The association of *A. muciniphila* with the mucin niche prompted us to evaluate fucosidase specificities on complex mucin *O*-glycans. We used a mixture of purified porcine gastric mucin (PGM), porcine colonic mucin (PCM), and bovine fetuin (BF). Despite differences as compared to human intestinal mucins, this substrate combination is powerful due to the large (160 structures) *O*-glycan diversity (Supplementary Table 3) and the presence of dense sulphation and sialylation in PCM similarly to the human counterpart. We evaluated activity on blood group A, H types 1-3, and four Le epitopes (Fig. 1a, b). No reliable activity of *Am*GH29A and *Am*GH29B was observed on the analyzed glycans. By contrast, *Am*GH29C and *Am*GH29D were active on all Le, but not H-epitopes (Fig. 1a, b). Both enzymes could accommodate single fucosylated Le[x] and Le[a] motifs, as well as double fucosylation (Le[b/y]) (Fig. 1a, b; Supplementary Fig. 5a–c). *Am*GH29D displayed a lower overall de-fucosylation yield than *Am*GH29C on single (Le[x]), double (Le[y]), or bifurcated fucosylated-epitopes, as well as cores 1, 2, 3, and 4 structures (Fig. 1a, b, Supplementary Fig. 5a–f). The lack of activity of *Am*GH29D on internal epitopes (Fig. 1c) is consistent with its observed lower overall defucosylation efficacy (Supplementary Table 4).

Both *Am*GH29C and *Am*GH29D were active on Le epitopes with a sialylated adjacent branch (Fig. 1d, e). In addition, a single sulphation at either the Gal or GlcNAc of Le[x] epitopes is tolerated (Fig. 1f, g), indicating that mono-sulphation is not restricting de-fucosylation. Notably, only *Am*GH29C, but not *Am*GH29D, showed activity on the double sulfated terminal Le[x] epitope (Fig. 1h) highlighting the overall broader epitope specificity of *Am*GH29C.

*Am*GH95A and *Am*GH95B share the α1,2-fucosidase activity (Supplementary Fig. 5g–j). However, marked differences in their epitope specificity and tolerance to non-fucosyl substitutions were observed (Fig. 1b). *Am*GH95A is specific for the H2, but lacks activity on H1 and H3 epitopes (Fig. 1i–k). The activity of *Am*GH95A is also impaired by double fucosylation, *e.g.* in Le[y] (Fig. 1l) and sulphation (Fig. 1m). The exclusive H type 2-specificity of *Am*GH95A is unprecedented amongst hitherto described fucosidases.

By contrast, *Am*GH95B has high activity on all H-type epitopes, double fucosylated Le[y] structures, sulfated H2 epitopes (Fig. 1i–m), and Le[y] epitopes at sulfated (Fig. 1o) or sialylated branches at core structures (Fig. 1p). Both enzymes were sensitive to non-reducing end substitution of the H-antigen (Fig. 1n) and were inactive on Fucα1,6-linked core of *N*-glycans (Supplementary Fig. 5k).

To unambiguously confirm enzyme regioselectivities on mucin-type glycoproteins, we harnessed glyco-engineered CHO cells, which display defined Lewis epitopes. *Am*GH29C and *Am*GH29D showed activity on conjugated Fucα1,3/4 linkages in Le[a/x] and Le[b/y] (Supplementary Fig. 6a–d), while *Am*GH95B hydrolyzed Fucα1,2 linkages in conjugated Le[b/y] epitopes (Supplementary Fig. 6c, d), which concurred with our MS-based analyses on mucin.

In summary, *Am*GH29C and *Am*GH95B resulted in the highest overall reduction of α1,3/4- and α1,2-fucosylation, respectively (Supplementary Table 4) and the highest relative activities on HMOs and

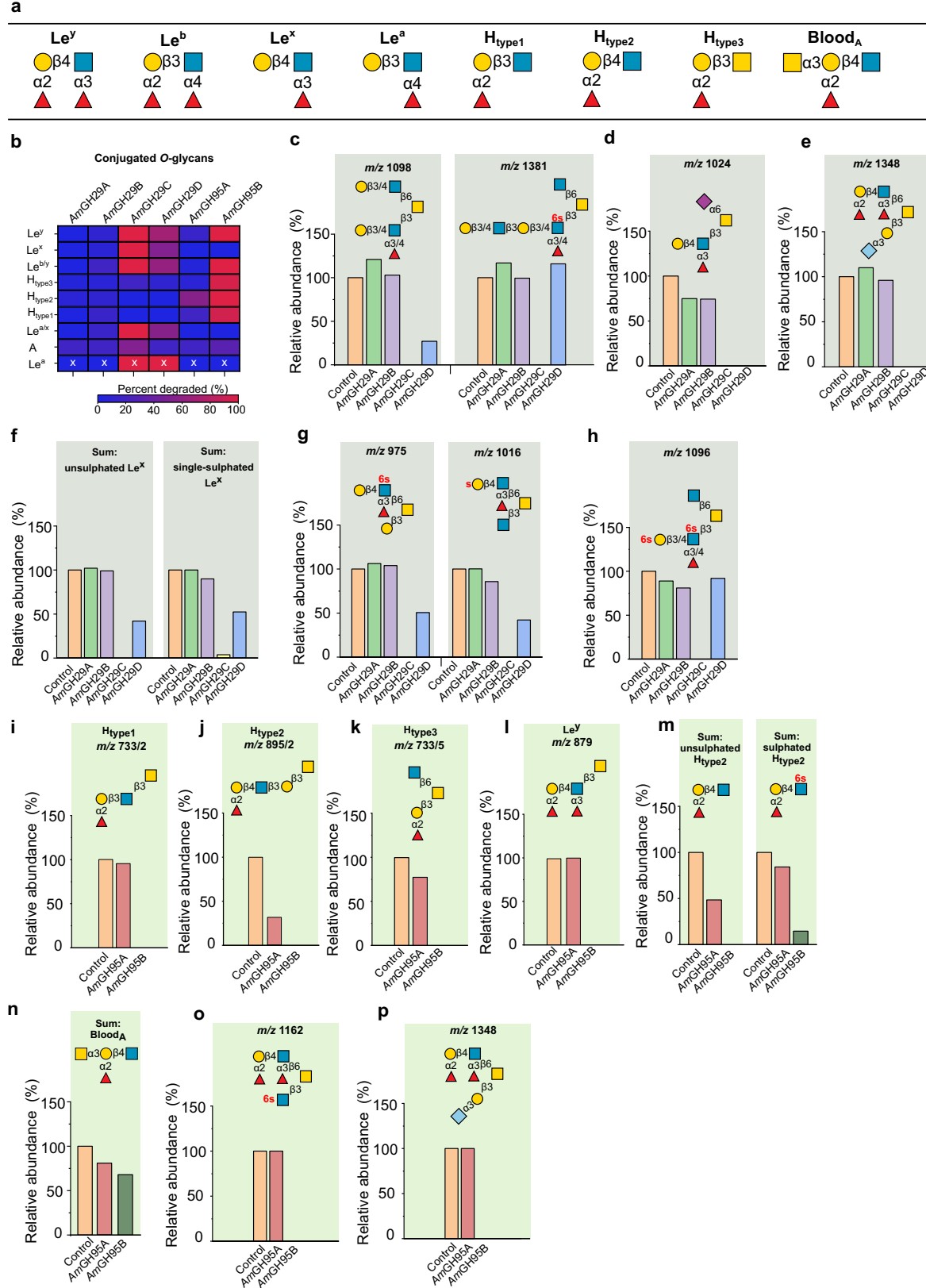

mucins (Supplementary Table 5). The broad specificity of *Am*GH29C is illustrated by activity on internal fucosylation and double sulphation, whereas *Am*GH95B was distinguished by activity on all H epitopes and double fucosylation. Collectively, the fucosidase suite allows full removal of mucin fucosyl substituents including from highly sulfated motifs.

## *A. muciniphila* sialidases remove all known mucin sialic acid linkages and include a sialyl-T-antigen-specific enzyme

The *Am*GH33A (Amuc_0625) and *Am*GH33B (Amuc_1835) are assigned into the CAZy family GH33. Two additional sequences (Amuc_0623 and Amuc_1547), with bacterial-neuraminidase-repeat-like domains that form the catalytic β-propeller fold in GH33 sialidases (Supplementary

**Fig. 1 | Activity profiles of the *A. muciniphila* fucosidases on mucin *O*-glycans. a** Overview of the fucosylated epitopes present in the analyzed conjugated *O*-glycans from porcine gastric and colonic mucins as well as fetuin. **b** The fucosidase activity heat map on different epitopes. **c** An example of enzyme sensitivity toward the fucosylation position in the glycan chain. **d**–**h** Examples illustrating the impact of sulphation (red lower case "s") and sialyl substitutions on fucosidase activity. **i**–**p** Examples of activity differences of the α1,2-fucosidases. Enzymes were incubated with mucin substrates blotted on membranes for 24 h, then *O*-glycans from enzyme-treated and non-treated controls were released and analysed. Relative abundances were calculated by integration of the LC-ESI/MS ion chromatogram

area under the curve (AUC) of each glycan peak normalized to the total. Relative abundances are also depicted in the activity heat map in panel b. Data are from a single experiment. The "x" marked data are obtained from a single glycan structure, due to the low abundance of assigned Le^a epitopes based on the MS data. Linkages of the sulphatyl substituent are only given if assignment was feasible. For isobaric (same m/z) glycans the additional number (/n) is the corresponding structure in the LC-ESI/MS data (Supplementary Data File 1). The slightly higher relative abundances compared to the controls reflect the noise due to minor variations in the amount of mucin used in each incubation.

Fig. 7a), were previously reported as active sialidases, based on an indirect chromogenic assay[25]. Sequences of Amuc_1547 and its close orthologues populate a separate and distant cluster in the GH33 phylogenetic tree (Supplementary Fig. 7b, c). Based on this as well as structural and mechanistic difference to GH33 sialidases, described below, the enzyme encoded by Amuc_1547 (*Am*GH181) is proposed as the defining member of the CAZy family GH181.

We expressed all four enzymes and assayed their activity on 3′-sialyl lactose (3′SL), 6′-sialyl lactose (6′SL), sialyl-Le^a, and α2,8-sialyl oligomers, which revealed activity for *Am*GH33A, *Am*GH33B and *Am*GH181, but not Amuc_0623 (Supplementary Fig. 8).

Next, we evaluated the four *A. muciniphila* sialidases against released *N*-glycans from human immunoglobulin G, released *O*-glycans and from PCM, and intact MUC2 from mouse (MUC2_Mouse), which has a dominant terminal Sd^a epitope[26]. Sialidase activity was observed for *Am*GH33A, *Am*GH33B, and *Am*GH181 on four abundant motifs, but Amuc_0623 activity was not measured (Fig. 2a–c). The active enzymes released both Neu5Ac and the animal-derived *N*-glycolylneuraminic acid (Neu5Gc), from porcine mucin (Fig. 2d, e). The motifs targeted by each enzyme were independent of the *O*-glycan core (Supplementary Fig. 9a–d).

Both GH33 enzymes were active on α2,3- and α2,6-sialyl linkages on both *O*- and *N*-glycans (Fig. 2b–e; Supplementary Fig. 9e). Striking specificity differences, however, between the two enzymes were observed. Thus, *Am*GH33B was not hindered by the substitution of the galactosyl moiety of Neu5Acα2,3Gal, *e.g.* on Sd^a_Core1, as opposed to *Am*GH33A which was inactive on this motif in released PCM *O*-glycans after 1 h incubation (Fig. 2a, b, f). By contrast, only *Am*GH33A displayed similar efficiency towards the extended sialyl-Tn epitope on both PCM-released *O*-glycans and MUC2_Mouse attached *O*-glycans after 1 h reactions (Fig. 2a–c, g). Similarly, only *Am*GH33A cleaved Neu5Acα2,3 Gal in both released PCM *O*-glycans and intact MUC2_Mouse after 1 h, whereas *Am*GH33B was inactive on the intact mucin substrate (Fig. 2b, c, Supplementary Fig. 9f). Our findings illustrate the importance of the sialyl density and glycan context (free/attached) in interrogating enzyme efficiencies on specific *O*-glycan motifs (Supplementary Tables 6 and 7), and merits caution when inferring enzyme specificities based on a single time point on free *O*-glycans.

Uniquely, *Am*GH181 displayed exclusive specificity towards the sialyl-T-antigen amongst the tested *O*-glycans, which was not hindered by α2,6-sialyation or β1,6-substitution of the GalNAc unit (Fig. 2a–c, e, g; Supplementary Fig. 9g). Thus, *Am*GH181 is inactive on a substituted Gal unit of the T-antigen, *e.g.* Sd^a epitopes (Fig. 2b, c, f), or on a different linkage/monosaccharides to the reducing end of the Neu5Ac α2,3Gal motif, *e.g.* in 3SLN (Fig. 2b, c, Supplementary Fig. 9f). To our knowledge, this strict specificity is unprecedented amongst reported sialidases.

### *Am*GH29D adopts a Cobra strike pose architecture, previously not observed in fucosidases

Both of the *Am*GH29C/D enzymes consist of a catalytic N-terminal domain, followed by a predicted galactose binding-like domain (GBLD), an unassigned sequence patch, and a C-terminal CBM32 (Supplementary Fig. 1a, Supplementary Fig. 2a, b). Amongst

biochemically characterized enzymes, this architecture was only observed in an orthologue from the mucolytic specialist *Bifidobacterium bifidum*. Although crystallization attempts of both *Am*GH29C/D were carried out, we could only determine the structure of *Am*GH29D, the most complex fucosidase structure to date (Fig. 3a, b). Unique to this structure is that the GBLD domain is joined to a linker domain and a C-terminal CBM32 (Fig. 3a, b, Supplementary Fig. 10a). The linker domain and the CBM32 adopt an extended conformation, which positions the CBM32 binding site above the catalytic domain (Fig. 3a, b). This juxtapositioning of the linker-CBM32 domain relative to the catalytic domain, which resembles a Cobra strike pose, is observed in the GH33 sialidase from *Micromonospora viridifaciens* (Supplementary Fig. 10a–c), suggesting convergent evolution to a similar putative mucin-binding motif. The GBLD and the CBM32 assume the same fold (Supplementary Fig. 10d), despite less than 22% shared sequence identity (Supplementary Fig. 1b). The position and the surface chemistry of the putative binding sites of the GBLD and the CBM32 domains are different, suggesting their possible functional divergence (Supplementary Fig. 10e–g). The catalytic site of *Am*GH29D is similar to the closest characterised counterpart from *Streptococcus pneumonia* (Supplementary Table 8), but differs by being flanked with a flat positively charged surface (Supplementary Fig. 11a), compatible with binding sialylated or sulfated glycans at the mucin surface.

To explain the specificity differences between *Am*GH29C and *Am*GH29D, we compared an AlphaFold model of *Am*GH29C to *Am*GH29D. Strikingly, the AlphaFold model of *Am*GH29C assumed a Cobra-bite pose with a rigid body bending of the CBM32-linker domains towards the active site, which is likely facilitated by the higher potential flexibility of the loop preceding the linker domains in *Am*GH29C than *Am*GH29D. (Supplementary Fig. 11d). Notably, shorter and more flexible loops result in a more open active site in *Am*GH29C as compared to *Am*GH29D (Supplementary Fig. 11d–g). This may promote the accommodation/recognition of bulky sulfated or larger fucosylated substrates, which are extended at the non-reducing ends, consistent with the activity of only *Am*GH29C on internal fucosylated GlcNAc (Fig. 1a–c, f; Supplementary Fig. 4k, n).

### Structural signatures of the inverting mechanism and strict specificity of *Am*GH181

To explain the strict specificity, we determined three structures of *Am*GH181: in ligand free form, bound with the transition-state inhibitor DANA (*N*-Acetyl-2,3-dehydro-2-deoxyneuraminic acid), and bound to both the T-antigen disaccharide (GNB) and DANA, which provides a mimic for a transition state-like substrate complex. The structure comprises an N-terminal catalytic domain joint to a C-terminal CBM-like domain (Fig. 3c, d). An inserted B domain between β-strands 1 (sheet I) and 2 (sheet II) of the CBM-like domain acts as a bridge by packing onto the catalytic domain via an extensive interface. A long loop in propeller blade 2 forms a Ca^2+-binding domain (Fig. 3c, d). The CBM-like domain occurs uniquely within GH181 (Supplementary Fig. 12a), with only very distant structural similarities to Galectin galactoside-binding domains (Supplementary Table 10).

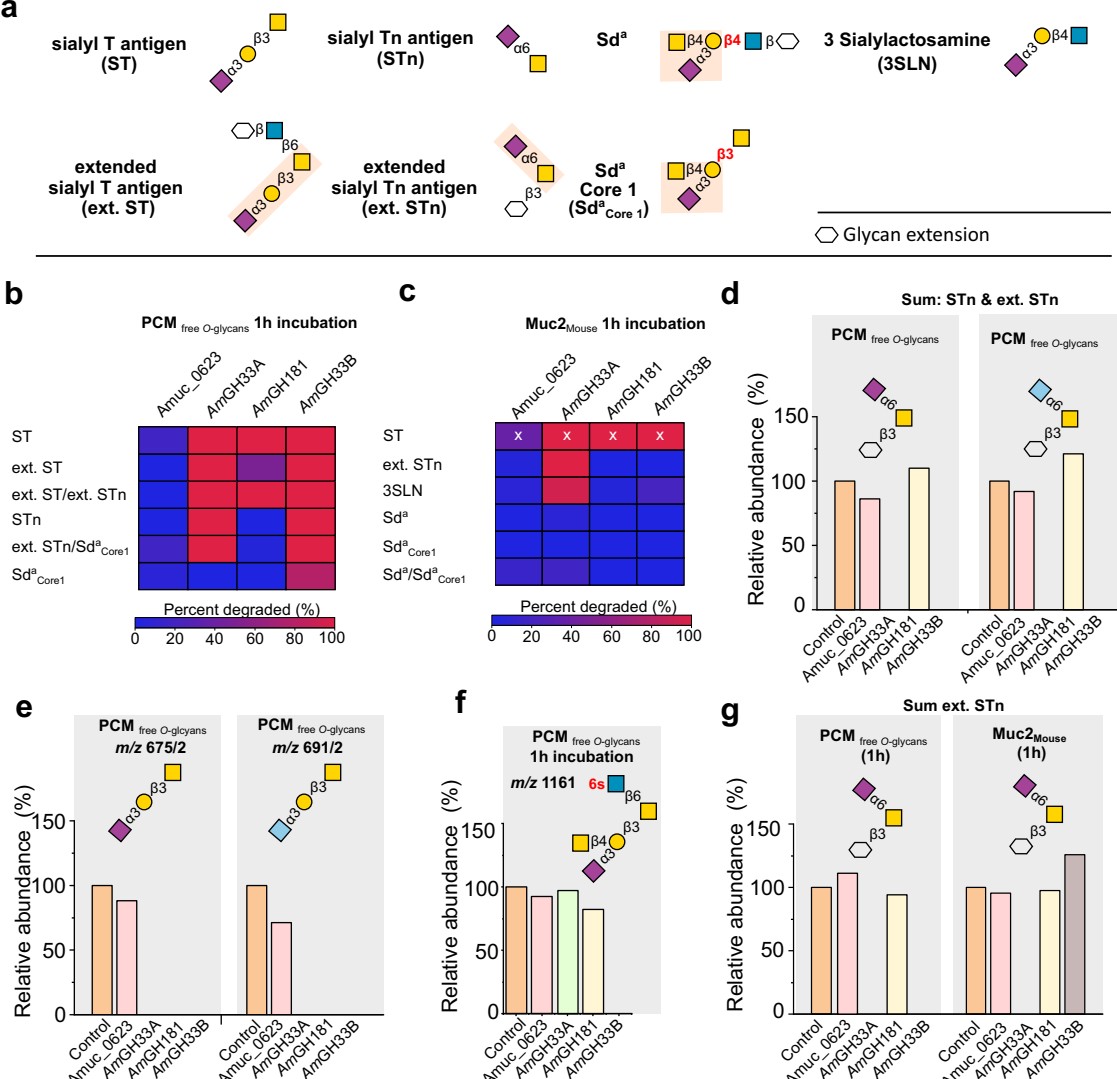

**Fig. 2 | The activity profiles of *A. muciniphila* sialidases. a** Overview of sialylated (denoted with "S") epitopes, in the analyzed mucin and released mucin *O*-glycans. **b**, **c** Activity heat maps on sialylated epitopes in released porcine colonic mucin *O*-glycans (PCM_free *O*-glcyans) and mouse MUC2 attached *O*-glycans (MUC2_Mouse), respectively after 1 h incubation. **d**, **e** Examples showing the broad substrate recognition of *Am*GH33A and *Am*GH33B towards α2,3- and α2,6-linked the Neu5Ac and Neu5Gc (light blue) sialic acid forms. **f** An example illustrating specificity differences between the investigated sialidases. **g** A comparison of activity profiles on the sum of sialylated extended Tn epitopes either in PCM_free *O*-glcyans and MUC2_Mouse. The enzymatic reactions were performed with PCM_free *O*-glcyans or MUC2_Mouse for 1 h and 24 h. The released PCM *O*-glycans were compared to non-treated controls, whereas MUC2 *O*-glycans were released after the enzymatic treatment and compared to non-treated controls. The relative abundances were calculated by integration of the LC-ESI/MS ion chromatogram peak of each glycan and normalising it to the total. The slightly higher relative abundances than the control in some incubations reflect the noise due to small variations in the mucin blots. The relative abundances are the basis for the activity heat map in panel. The "x" marked data are obtained from a single glycan structure, due to the low abundance of the sialyl T epitope in the MUC2_Mouse sample. Sulphatyl substitutions are denoted with "s" in red.

The catalytic module of *Am*GH181 is distantly related to counterparts in GH33 (Supplementary Table 9). The catalytic site comprises a shallow pocket, flanked by a positive electrostatic potential (Supplementary Fig. 13a). The active site is open at one side of the β-propeller due to shorter loops as compared to GH33 enzymes. The $Ca^{2+}$-binding domain, the B domain, and two large loops pack onto the β-propeller to give the enzyme a "sun-chair" architecture (Fig. 3c, d; Supplementary Fig. 13b–f).

At the catalytic site, R234 and R305 are shared with GH33, whereas glutamine (Q367) substitutes the third arginine in the GH33 conserved triad (Fig. 3e, f; Supplementary Fig. 14a).

A unique signature is the substitution of the tyrosine catalytic nucleophile in GH33 members to a glutamine (Q350) that is preceded by a histidine (H349) in *Am*GH181 (Supplementary Fig. 14b–d). Strikingly, Q350 and an adjacent glutamate (E218), both invariant in GH181

(Supplementary Fig. 12b, c), are potentially hydrogen bonded to a water molecule that overlays with the oxygen in the catalytic tyrosine of GH33 enzymes. This water is positioned for nucleophilic attack at the C2 of the sialyl (or the inhibitor) at subsite −1 (Fig. 1e, f; Supplementary Fig. 14b). A solvent tunnel connects the bulk of the solvent to the enzyme catalytic site (Supplementary Fig. 14e, f), similarly to some exoglycosidases[27]. An invariant aspartate (D345), unique for GH181, is hydrogen bonded to the Gal C3-OH group at subsite +1 (Supplementary Fig. 14b). Based on these data, we hypothesized that *Am*GH181 adopts an inverting mechanism that involves nucleophilic attack by the activated water molecule that overlays with the oxygen of the catalytic tyrosine in GH33 enzymes. To investigate the stereochemical mechanism, real-time NMR spectroscopy was used to monitor the hydrolysis time course of 3´-sialyllactose by *Am*GH181 and of 6´-sialyllactose by *Am*GH33A as a control. The initial emerging signals were

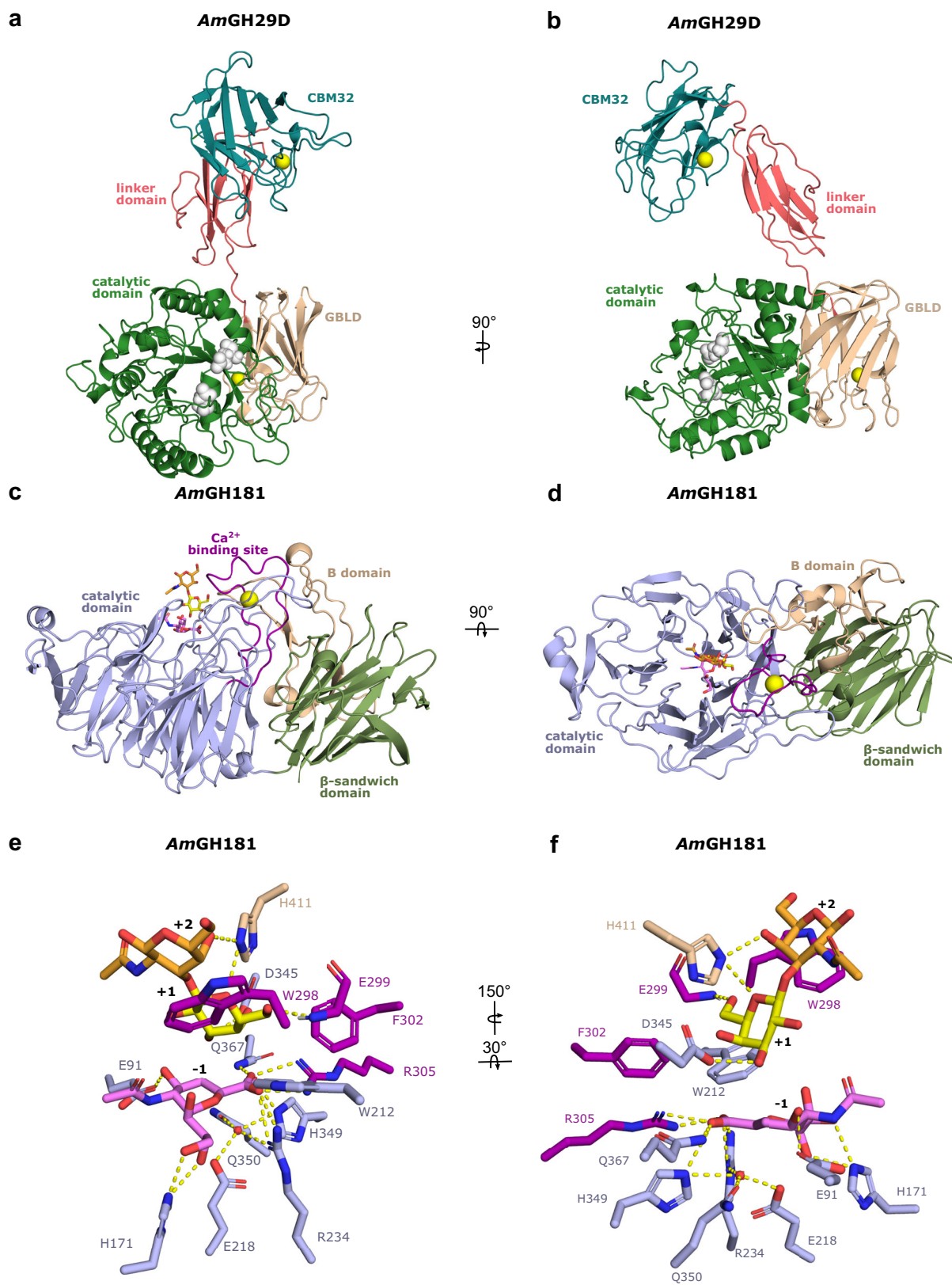

for β-Neu5Ac and α-Neu5Ac, in the reactions catalyzed by *Am*GH181 and *Am*GH33A, respectively (Supplementary Fig. 15). These data corroborated the proposed inverting mechanism of *Am*GH181, which is unprecedented in the evolutionary related CAZy clan E that includes GH33 retaining sialidases. Of note, the defining member of GH156 is the only other reported inverting exo-sialidase[28] outside clan E, while viral endo-sialidases of GH58 are also inverting[29].

Comparison of the ligand-free and the ligand-bound structures of *Am*GH181 revealed the flipping of a tryptophan (W298, 56.3 % conserved within GH181, Supplementary Fig. 12c) in the ligand-bound structures. This conformational change positions W298 (Supplementary Fig. 16a, b) to stack onto the GalNAc unit of the T-antigen, thereby defining subsite +2. The invariant histidine (H411) from domain B forms potential hydrogen bonds to the Gal and GalNAc units. Thus, W298

**Fig. 3 | The crystal structures of the *Am*GH29D fucosidase and *Am*GH181 sialidase from *A. muciniphila*. a** Overall structure of *Am*GH29D comprising a catalytic N-terminal (β/α)₈ domain (amino acids 38-362), a predicted galactose binding like domain (GBLD, aa 363-489), a linker domain comprising two β-sheets formed by five antiparallel strands (aa 490-571) and a C-terminal CBM32 (aa 572-704). The inferred catalytic residues (white) and the bound Ca²⁺ ions (orange) are shown as spheres. **b** A 90° rotation of the view in a. **c** Overall structure of *Am*GH181, comprising an N-terminal 6-fold β-propeller catalytic domain (aa 23-281 and 307-384) with a Ca²⁺-binding domain formed by an extended loop between two inner β-strands in propeller blade 2 (aa 282-306). The Ca²⁺ (orange sphere) is assigned based on coordination geometry and distance. The catalytic domain is joint to a C-terminal β-sandwich CBM-like domain (residues 457-595) and an inserted B domain (aa 391-456) between the β-strands 1 (sheet I) and 2 (sheet II) of the CBM-like domain. **d** A 90° rotation of the view in c. **e, f** The active site of *Am*GH181 with the DANA inhibitor (subsite −1) and the T-antigen disaccharide bound at the +1 and +2 subsites in two different orientations. The same domain colours are used in panels c–f.

and H390 form a "sugar tong" that restricts the T-antigen disaccharide. The galactosyl at subsite +1 stacks onto an invariant tryptophan (W212) and is additionally recognized by two hydrogen bonds (Fig. 3e, f). Collectively, these aromatic stacking and polar interactions provide a plausible explanation for the strict specificity of the enzyme.

The presence of two potential saccharide surface binding sites (SBSs) is intriguing. The first site is adjacent to the active site, where an inhibitor molecule was modelled, whereas a galactose unit was modelled at the second binding site located on the opposite side of the catalytic site (Supplementary Fig. 16c–g). Although both SBSs are conserved in *Akkermansia* sequences that populate a single clade in the phylogenetic tree of GH181, only moderate conservation was observed for other phylogenetic clusters (Supplementary Fig. 12d–g). The presence of a CBM-like domain and of two potential secondary binding sites are indicative of the association of the enzyme to mucin, which is explored below.

### Fucosidases and sialidases display mucin binding and their corresponding activities are extracellular

The presence of putative CBMs prompted us to investigate enzyme binding to PGM. Strikingly, *Am*GH29C, *Am*GH29D, *Am*GH33B, and *Am*GH181, all containing annotated or putative CBMs, were mainly bound to mucin in pull-down assays (Supplementary Fig. 17a, c, d), whereas no binding or weak binding (*e.g.* GH95 enzymes, Supplementary Fig. 17b, d) was observed for the rest of the enzymes.

To test localization, we grew cells on PCM and assayed the intact cells, culture supernatants, and the cell lysates against 2´FL (GH95 substrate), Leᵃ trisaccharide (*Am*GH29C/*Am*GH29D substrate), the disaccharide Fucα1,3GlcNAc (*Am*GH29A/*Am*GH29B substrate) and 6´SL (*Am*GH33A/*Am*GH33B substrate) as well as 3´SL that is a poor substrate for both the GH33 enzymes and *Am*GH181. Enzymatic activity on 2´FL, Leᵃ, but not Fucα1,3GlcNAc, was detected mainly in the cell fraction (Supplementary Fig. 18a–j), which is consistent with the cell-attachment of the fucosidases. The activities against 6´SL and 3´SL were shown for the intact cell fraction and the supernatant (Supplementary Fig. 18k–r), which suggests that at least one sialidase is cell-attached and at least one is secreted.

### *A. muciniphila* fucosidases and sialidases are crucial for growth on mucin and sharing of mucin derived sugars

Lacking tools for generating gene knock-outs in *A. muciniphila*, we deployed inhibitors to probe the impact of the fucosidases and sialidases on mucin growth. First, we evaluated the inhibition potency toward the fucosidases (IC₅₀ < 54 μM) and sialidases (IC₅₀ < 200 μM) (Supplementary Tables 11 and 12). *A. muciniphila* cells grew undistinguishably on monosaccharides in presence or absence of 1 mM or 20 mM (Fig. 4a, b) of each inhibitor. By contrast, growth on mucin (PCM) in the presence of the inhibitors was heavily impaired at 1 mM and abolished at 20 mM of each inhibitor (Fig. 4a, b; Supplementary Table 13). The results showed that the fucosidase and sialidase suite was critical for initiating growth on mucin. Next, we compared the growth of *A. muciniphila* on the intact and de-sialylated/de-fucosylated PCM. The difference in growth profiles was modest (Fig. 4c), indicating that the removal of fucose/sialic acid was not the limiting step during growth on mucin. Growth assays showed that sialic acid did not sustain

*A. muciniphila* growth, whereas fucose supported very slow growth, in agreement with previous studies[30,31] (Fig. 4d). The lack of relevant contribution of the released sialic acid and fucose during growth on mucin highlighted possible nutritional sharing of these monosaccharides with other microbiota members. We co-cultured *A. muciniphila* and mucus-associated model butyrogenic Clostridia to investigate potential syntrophy. The tested strains grew insignificantly on mucin, consistent with low butyrate levels in culture supernatants (Fig. 4e–n). Higher butyrate concentrations, however, were observed in co-culture supernatants of especially *Roseburia inulinivorans* and *Faecalibacterium prausnitzii*, in excellent agreement with the growth of both species on sialic acid (Supplementary Fig. 19a), whereas lower or no butyrate production was observed for the remaining strains (Fig. 4f, h, j, i, and n). The poor growth on decapped mucin suggested that the tested Clostridia are not efficiently competing with *A. muciniphila*, which was in agreement with the dominance of *A. muciniphila* and low relative abundance of the Clostridia based on qPCR analyses (Supplementary Fig. 19b–f). Sialic acid utilisers showed the highest relative abundances with about 9% in the *A. muciniphila* co-culture with *Faecalibacterium prausnitzii*, while *Roseburia faecis* and *Agathobacter rectalis* were not detectable after 24 h in the co-cultures. We also looked at the impact of sialic acid on butyrate production in co-cultures using low sialylated PGM (0.5–1.5% w/w sialic acid) and the highly sialylated bovine submaxillary mucin (BSM, 9-24.5% w/w sialic acid). The largest increase in butyrate production in co-cultures was observed with the sialic acid-utilising strains on the densely sialylated BSM, whereas differences on PGM were less pronounced (Supplementary Fig. 20a–d). Additionally, the degree of mucin sialylation had a minor impact on the butyrate production by the strains that do not utilise sialic acid in co-culture with *A. muciniphila* (Supplementary Fig. 20e–h). Collectively, these data indicate that sialic acid sharing appears to be the main driver behind the increased butyrogensis in the co-cultures. Our findings suggest that the decapping enzymes promote direct sharing of their product monosaccharides to notably abundant butyrate producers, besides their key role in granting access to the underlying glycans and initiating growth on mucin.

## Discussion

The modulation of the metabolic and immune systems of the host by *A. muciniphila*[32,33] correlates to the relative abundance of this symbiont and its interplay with other mucus-associated bacteria. The molecular mechanisms of mucin *O*-glycan degradation[34] and utilization by *A. muciniphila* remain largely underexplored. Here, we present detailed enzymatic, microbiological and structural analyses, which led to the identification of key glycoside hydrolases that collectively were able to removal all known mucin fucose and sialic acid caps, from mucin *O*-glycans.

The high prevalence of the genes encoding the most active fucosidases and sialidases in different *A. muciniphila* strains (Supplementary Table 14), underscores the importance of these enzymes for mucin *O*-glycan utilization. In addition, the sequence divergence of catalytic modules is consistent with the observed distinct enzymatic signature of each enzyme.

Interestingly, two of the enzymes, *Am*GH95A and *Am*GH181, display specificities towards a single glycan each, *e.g.* the abundant H2[35]

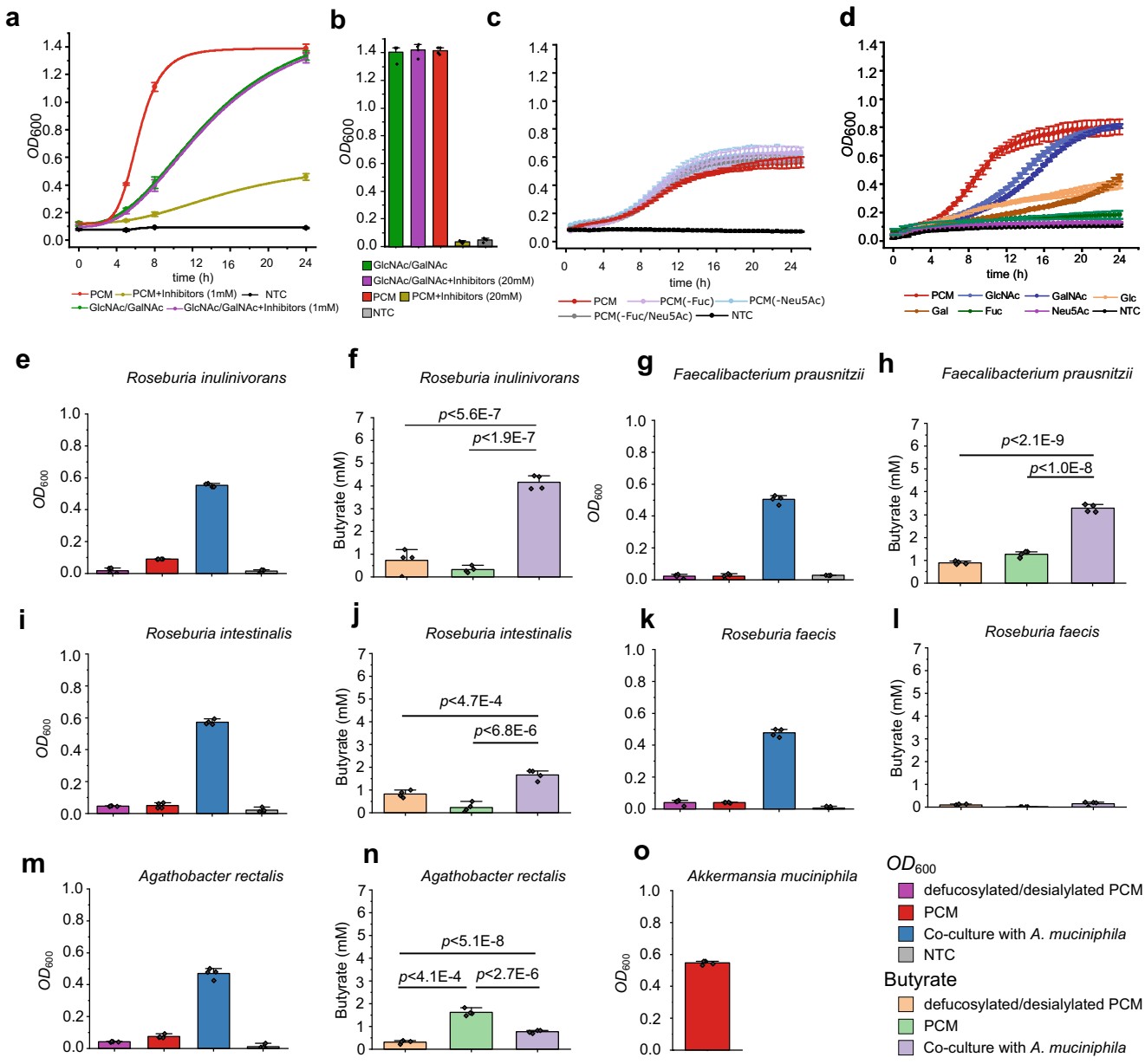

**Fig. 4 | Fucosidases and sialidases of *A. muciniphila* are critical for growth on mucin and contribute to nutrient sharing with butyrate-producing Clostridia.** **a** Growth curves of *A. muciniphila* on porcine colonic mucin (PCM) and on an equimolar mixture of GlcNAc/GalNAc, alone or in the presence of a equimolar mixture of fucosidase (DFJ, 1-Deoxyfuconojirimycin) and sialidase (DANA) inhibitors (1 mM each) as compared to no-carbon source control (NTC). **b** Same as in a, but *A. muciniphila* growth is shown after 24 h in the presence or absence of 20 mM of each inhibitor. **c** Growth curves of *A. muciniphila* on PCM, de-fucosylated PCM and de-sialylated PCM compared to a no-carbon source control. The growth experiment was performed in a microtitre plate as opposed to panel a, where the growth ($OD_{600}$) was measured in cuvettes. **d** Growth curves of *A. muciniphila* on monosaccharides from mucin and a no-carbon source control. **e**, **g**, **i**, **k**, **m** Growth levels of different butyrate producing Clostridia from the Oscillospiraceae and Lachnospiraceae families in microtitre plates on PCM and de-fucosylated/de-sialylated PCM (including a no-carbon source control) in monocultures or in co-culture with *A. muciniphila* on PCM after 24 h. **f**, **h**, **j**, **l**, and **n**. The corresponding butyrate concentrations in culture supernatants. **o** Growth level of *A. muciniphila* in monoculture on PCM after 24 h. Growth analyses (**a**–**d**, **e**, **g**, **i**, **k**, **m** and **o**) on media supplemented with 0.5% (w/v) mucin or a 1:1 mix of GlcNAc and GalNAc were performed in four independent biological replicates (*n* = 4) and butyrate in culture supernatants was analyzed from 3 independent growth experiments (*n* = 3). The growth data and butyrate quantifications are mean values with standard deviations (SD) as error bars. The statistical significance between butyrate concentrations reached was evaluated using an unpaired two-tailed Student's *t*-test and the corresponding *p*-values are included in the figure panels. Source data are provided as a Source Data file.

and the sialyl-T-antigen epitope, respectively. To our knowledge, *Am*GH95A and *Am*GH181, are the most specific fucosidase and sialidase reported to date. By comparison, *Am*GH95B and *Am*GH29C are highly promiscuous, enabling the removal of multiple complex epitopes, *e.g.* *Am*GH95B targets H1, H2, H3 type and Le[b/y] antigens (Fig. 1b, i–m, o, and p) and *Am*GH29C is active on internal fucosylations, sialylated Le[a/x] and Le[b/y] epitopes as well as double sulfated motifs (Fig. 1b–h,

Supplementary Table 4). The previously not reported ability to hydrolyze fucose from double sulfated motifs suggests that de-fucosylation by *A. muciniphila* is feasible without prior de-sulphation.

The evolution of a specificity gradient that spans mono-epitope-specific to highly promiscuous fucosidases and sialidases, may be driven by the optimization of enzyme affinities ($K_m$) to promote efficient decapping of highly diverse mucin *O*-glycans to promote growth.

Thus, *A. muciniphila* deploys "bulldozer" enzymes possessing open active sites to accommodate and rapidly decap multiple bulky complex glycans as described above. Such enzymes are unlikely to have sufficient affinity for simpler common mucin epitopes, *e.g.* the H2 and sialylated T and Tn antigens. High selectivity and catalytic efficiency require specialized enzymes with restricted active sites to recognize specific simple epitopes with favourable binding energy. The importance of enzyme affinity is also evident from the strong association of *Am*GH29C, *Am*GH29D, *Am*GH33B, and *Am*GH181 to mucin, which correlates to the presence of putative CBMs and saccharide surface binding sites, both known to increase the enzymatic efficiency of CAZymes towards insoluble substrates[36,37].

The observed mainly cell-attached localization of fucosidase and detection of both cell-attached and secreted sialidase activities indicates that sialic acid and fucose removal occurs extracellularly, which is crucial for further mucin glycan breakdown by *A. muciniphila* exoglycosidases and endo-glycanases[34]. Strikingly, four of the recently described *O*-glycopeptidases from *A. muciniphila*, revealed a preference for either the T- or the Tn-epitopes, whereas the presence of a sialyl decoration abolished or severely impaired their activity[38–41]. These findings suggest that a key role of the sialidases is to create sites for *A. muciniphila O*-glycopeptidases to allow cleavage of the mucin backbone (Fig. 5). Recently, it has been proposed that *A. muciniphila* appears to internalize mucin fragments to the periplasmic space[42]. Since *A. muciniphila* uses the mucin backbone as a nitrogen source, it is highly likely that at least one or more of its *O*-glycopeptidases (mucinases) are also localised extracellularly to cleave the mucin backbone into fragments that are amenable to internalization. This strategy is consistent with the impairment of *A. muciniphila* growth on PCM in the presence of sialidase and fucosidase inhibitors (Fig. 4a, b) and growth defects of *A. muciniphila* mutants in the genes of the highly active sialidase *Am*GH33B and fucosidase (*Am*GH95B)[42]. Collectively, these data suggest a critical role of the decapping enzymes of *A. muciniphila* in initiating mucin breakdown.

*Bifidobacterium bifidum* extracellular fucosidases and sialidases mediate cross-feeding on mucin with other infant gut bifidobacteria[43]. Here we show the corresponding enzymes from *A. muciniphila* may confer a similar ecological role amongst the mucus-adherent microbial community in adults, which is dominated by Lachnospiraceae, Oscillospiraceae (Fig. 4d–m) and to a less extent *Bacteroides* species[44,45]. The syntrophy between *A. muciniphila* and sialic acid-utilising model butyrogenic Clostridia is likely to be beneficial to the host, as it partially nourishes this health-beneficial microbiota group, without competing with *A. muciniphila* or contributing to mucin breakdown. The levels of butyrate in the co-cultures with *A. muciniphila* correlated to the ability of co-cultured Clostridia to grow on sialic acid, which was proportional to the density of sialic acid decoration on the mucin substrate. Indeed, distinct Clostridia possess efficient ABC transporters for the uptake of sialic acid[46,47]. These results unveil the importance of this monosaccharide in the observed syntrophy with *A. muciniphila*[47,48]. The very poor growth of *A. muciniphila* on fucose (Fig. 4d) is also expected to allow sharing this monosaccharide with fucose-utilizing bacteria[49]. Further work is needed to map the broadness of fucose and sialic acid utilization amongst the mucus-adherent community.

In conclusion, microbial mucin turnover is of paramount significance for the maintenance of symbiosis between the host and the mucus-associated microbiota, as well as for pathologies linked to the excessive breakdown of the mucosal layer. Our findings offer unprecedented insight into the enzymatic apparatus that initiates growth on mucin and exposes glycopeptidase cleavage sites as well as promotes nutrient sharing by a key dedicated mucolytic symbiont with the mucus-associated microbiota. Further work, however, is needed to decipher the downstream steps in mucin breakdown by *A. muciniphila* and the complex trophic interactions of this specialist with the mucin-associated microbiota. The exquisite specificity of distinct *A. muciniphila* sialidases and fucosidases expands the analytical toolbox for unambiguous linkage assignment in MS-based *O*-glycan analyses or for targeting specific glycan motifs.

## Methods

The study conforms to the ethical guidelines articulated in the Danish code of conduct for research integrity (Danish Ministry of Higher Education and Science, ISBN: 978-87-93151-36-9). The animal material in the study was either commercially procured or prepared in previous studies that are cited.

### Chemicals and carbohydrates

*p*-Nitrophenyl α-L-fucopyranoside (*p*NPFuc), *N*-acetylneuraminic acid (Neu5Ac), α-L-fucose (Fuc), Galactose (Gal), Glucose (Glc), *N*-acetylglucosamine (GlcNAc), *N*-acetylgalactosamine (GalNAc), Fetuin (from fetal bovine serum), mucin from bovine submaxillary gland, and Type III porcine gastric mucin were from Sigma (St. Louis, MI, USA). Lewis antigens (Le$^a$ triose, Le$^x$ triose, Le$^b$ tetraose, Le$^y$ tetraose), 6-Sialyllactose (6′SL), 3-Sialyllactose (3′SL), 2′-Fucosyllactose (2′FL), 3-Fucosyllactose (3FL) were purchased from Dextra (Reading, UK). αFuc1,3GalNAc, αFuc1,4GalNAc, αFuc1,2Gal and αFuc1,3Gal were from Toronto Research Chemicals (Toronto, Canada), Lacto-*N*-difucopentaose I (LNDFP I), Lacto-*N*-difucopentaose II (LNDFP II), Lacto-*N*-difucohexaose I (LNDFH I), Lacto-*N*-difucohexaose II (LNDFH II), sialylated Le$^a$ triose (sialyl Le$^a$), Colominic acid, *N*-Acetyl-2,3-dehydro-2-deoxyneuraminic acid sodium salt (DANA), 1-Deoxyfuconojirimycin HCl (DFJ), 4-Methylumbelliferyl *N*-acetyl-α-D-neuraminic acid sodium salt (4MU-Neu5Ac) and Galacto-*N*-biose (GNB) were from Carbosynth (Berkshire, UK). Recombinant P-selectin glycoprotein ligand-1 (PSGL-1) with *O*-glycans that are terminated with Lewis antigens were prepared in Gothenburg University. Antibodies against Lewis antigens were from Sigma and Santa Cruz biotechnology. All purchased chemicals were of analytical grade unless otherwise stated and were used without further purification.

### Porcine gastric and colonic mucins and mouse colonic mucin

Type III porcine gastric mucin (PGM), purchased from Sigma (St. Louis, MI, USA), was further purified according to Miller et al.[50]. Briefly, 2.5% (w/v) mucin was dissolved in phosphate-buffered saline pH 7.2 and stirred (20 h, room temperature), followed by centrifugation (10,000 *g*, 30 min, 4 °C). The soluble mucin-containing supernatant was collected and precipitated by adding ice-cold EtOH to 60% (v/v) twice. Thereafter, the purified soluble mucin was dialyzed against Milli-Q using a 50 kDa molecular weight cut-off membrane (SpectraPore7, Rancho Dominguez, CA, USA), freeze-dried, and subsequently stored at −20 °C until further use. Porcine colonic mucin (PCM) and MUC2 from mouse (MUC2$_{Mouse}$) were prepared and purified as previously described[47,51].

### Preparation of *O*-glycan from PGM and PCM and *N*-glycans from human IgG

*O*-glycans from PCM and PGM were released from mucin glycoproteins by reductive amination before glycans were desalted and dried as previously described[52]. *N*-glycans were released from human serum IgG (Sigma-Aldrich, I4506) using PNGase F (CarboClip, Spain). In short, human IgG (1 mg) was reduced (10 mM DTT, 95 °C, 20 min) and alkylated (25 mM iodoacetamide, in dark at RT for 1 h). The *N*-glycans were then released by PNGase F in 50 mM NH$_4$OAc (pH 8.4), 37 °C overnight incubation, before reduction (0.5 M NaBH$_4$ in 20 mM NaOH, 50 °C overnight), desalting, and drying[52].

### Cloning, expression, and purification of putative fucosidases and sialidases

The gene fragments of the glycoside hydrolase families GH29, GH95, GH33, and the putative sialidases with Bacterial-Neuraminidase-Repeat

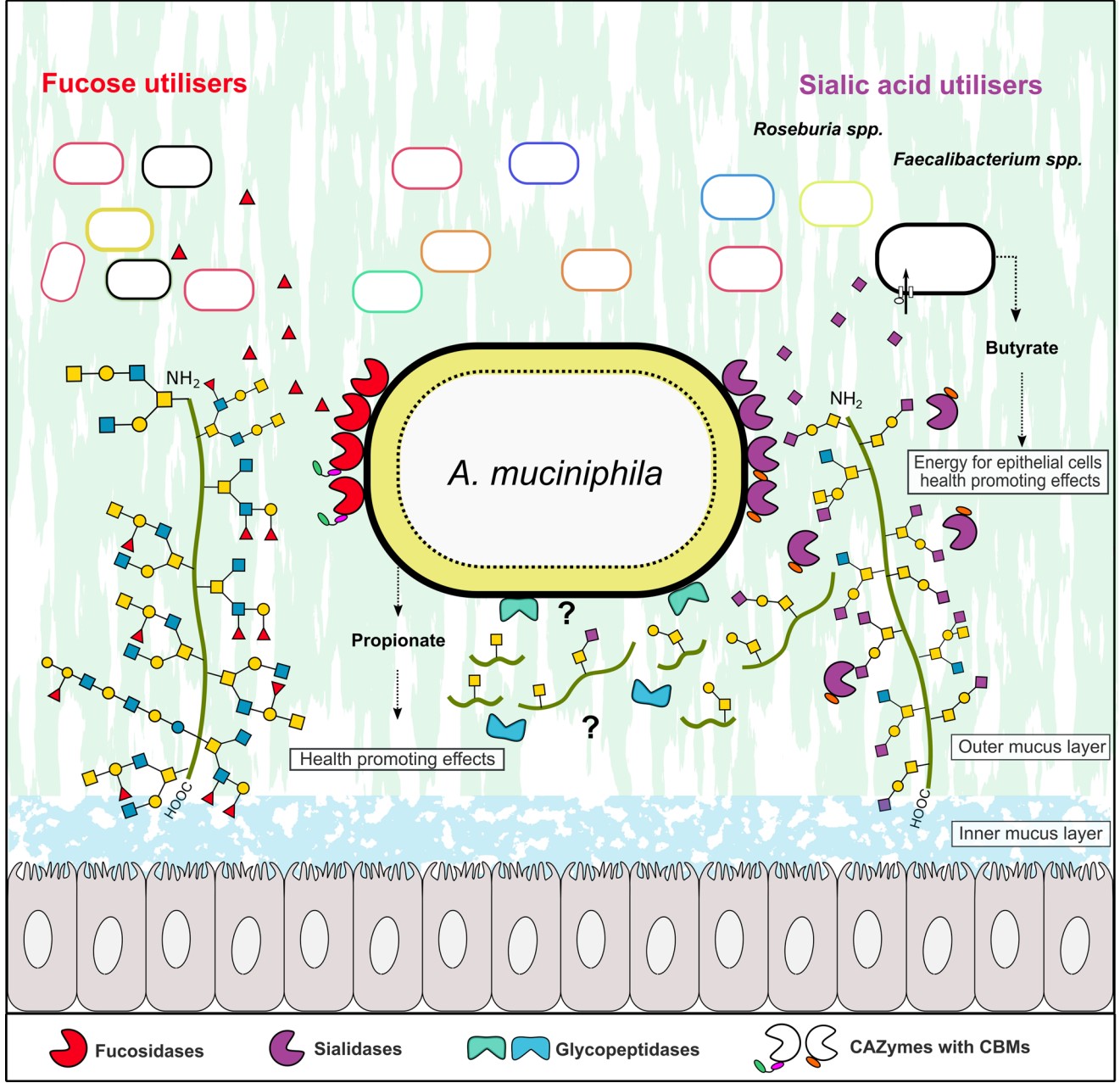

**Fig. 5 | Model for mucin defucosylation and desialylation by *A. muciniphila* and sharing of fucose and sialic acid with mucus adherent butyrogenic gut microbiota.** The Grey cells represent epithelial colonocytes, and the green line represents the mucin peptide backbone. The fucosidase activity was detected mainly in the intact *A. muciniphila* cell fraction, which is consistent with an extracellular cell-attached localization. Sialidase activity on 3´-sialyllactose and 6´-sialyllactose was detected both in intact cells and culture supernatant, which is consistent with the cell attachment of at least one sialidase and secretion of at least one sialidase. The removal of sialic acid exposes the recognition motifs for glycopeptidases that have been shown to cleave primarily adjacent to non-sialylated T and/or Tn epitopes, thereby enabling the cleavage of the mucin peptide backbone. The localization of the glycopeptidases has not been experimentally proven, but it is inferred, as intact mucin is too large to be internalized. The released sialic acid was not utilised by *A. muciniphila* and was shown to confer butyrate production by sialic-acid utilising model butyrogenic Clostridia. Similarly, fucose did confer meaningful growth of *A. muciniphila* (see Fig. 4d), making it available for cross-feeding to fucose utilisers. The figure background is inspired from figure 6 in ref. [47], cited in the present study.

(BNR)-like domains, which encode the mature peptides lacking the signal peptides (as predicted by SignalP 5.0)[53], were amplified from *Akkermansia muciniphila* ATCC BAA-835 (DSM 22959) genomic DNA using the primers as shown in (Supplementary Table 1). Infusion cloning (Clonetech/Takara, CA, USA) was used to clone these amplicons into the NcoI and XhoI sites of the pET28a(+) vector (Novagen, Madison, WI). The resulting recombinant plasmids, which encode the GH29, GH33, and GH95 enzymes were transformed into *Escherichia coli (E. coli)* DH5α and transformants were selected on LB plates

supplemented with kanamycin (50 µg mL⁻¹). After full sequencing, the plasmids were transformed into *E. coli* BL21 (DE3) Δ*lacZ* production strain (a kind gift from Professor Takane Katayama, Kyoto University, Kyoto, Japan) and the transformants were grown in 2 L LB medium with kanamycin (50 µg mL⁻¹) at 30 °C to $OD_{600} \approx 0.5$, followed by cooling on ice for 30 min before induction with isopropyl β-D-1-thiogalactopyranoside (IPTG) to 200 µM. Thereafter, growth was continued overnight at 18 °C and cells were harvested by centrifugation (10,000 g, 30 min), re-suspended in 10 mL of the purification

buffer A (20 mM HEPES, 500 mM NaCl, 10 mM imidazole, 10% (v/v) glycerol, pH 7.5) and disintegrated by one passage through a high-pressure homogenizer (SPCH-1, Stansted Fluid Power, Essex, UK) at 1000 bar, followed by incubation for 30 min on ice with 5 μl benzonase nuclease (Sigma). The lysates were then centrifuged for 20 min at 45,000 g and 4 °C and the supernatants were filtered (0.45 μM) and loaded onto HisTrap HP columns (GE Healthcare, Uppsala, Sweden). Then, bound proteins were washed (13 column volumes, CV) and eluted with the same buffer using an imidazole gradient from 10 to 400 mM in 15 CV. Pure protein fractions based on SDS-PAGE analysis were collected, concentrated, applied onto a HiLoad 16/600 Superdex 75 prep grade column (GE healthcare) and eluted by 1.2 column volumes of 20 mM HEPES, 150 mM NaCl, pH 6.8. The fractions containing each enzyme were pooled and concentrated (10 kDa Amicon® Ultra Centrifugal filters, Millipore, Darmstadt, Germany). Pure fractions, as judged by SDS-PAGE analysis, were pooled, concentrated as above and the protein concentration was determined using a Nanodrop (Thermo, Waltham, MA) using the theoretically predicted Molar extinction coefficients ($\varepsilon_{280}$) using the ProtParam tool (http://web.expasy.org/protparam). Finally, $NaN_3$ (0.005% w/v) was added to the enzyme stocks that were stored at 4 °C for further use.

### Enzyme activity and inhibition assays on synthetic substrates

All enzyme activity and inhibition reactions were performed in 20 mM HEPES, 150 mM NaCl, pH 6.8 unless otherwise state. For fucosidase kinetics, the initial reaction rates of AmGH29A (0.5 nM), AmGH29B (20 nM), AmGH29C (400 nM) AmGH29D (250 nM), AmGH95A (10 nM) and AmGH29B (50 nM) were determined on seven pNPFuc concentrations in the range 0.25–15 mM (except for AmGH95B, which was extended with a 30 mM substrate concentration). The reactions were carried out at 37 °C for 3 hours for all enzymes except AmGH29D, which was incubated for 4 hours. Aliquots of the reactions were collected at 30 min and 40 min intervals for 3 h and 4 h reactions, respectively, and quenched into $Na_2CO_3$ (0.4 M final concentration). The concentration of the pNP enzymatic product was determined by measuring $A_{405\,nm}$ using a 96-well plate reader (BMG Labtech, Ortenberg, Germany) using a pNP standard curve (0 to 140 μM pNP). The Michaelis-Menten equation was fit to the initial rates using Prism 6 (GraphPad San Diego, USA). For determining the inhibition constants ($IC_{50}$), reactions were performed continuously at 37 °C for 30 min in a microtiter plate and absorbance ($A_{405\,nm}$, fucosidases) or emission ($E_{450\,nm}$; $Excitation_{370\,nm}$, sialidases) was measured in 60 sec intervals using a 96-well plate reader. The initial reaction rates of AmGH95A (0.25 μM), AmGH95B (0.5 μM), AmGH29A (0.5 μM) AmGH29B (0.5 μM) AmGH29C (10 μM) and AmGH29D (10 μM) were determined using 2 mM pNPFuc and Deoxyfuconojirimycin (DFJ) over a concentrations range of 0.1–10 mM (AmGH29A, AmGH29B and AmGH29C) or 1–100 mM (AmGH95A, AmGH95B and AmGH29D). Sialidase inhibition reactions were determined at enzyme concentration of 50 nM, expect Amuc_0623 which was assayed at 200 nM. The initial rates were determined using 1 mM 4-Methylumbelliferyl N-acetyl-α-D-neuraminic acid (4MU-Neu5Ac) and 0.01–1 mM N-Acetyl-2,3-dehydro-2-deoxyneuraminic acid (DANA). A Hill equation was fit to the initial rates using OriginPro 2021. All enzyme activity and inhibition reactions were performed in independent triplicates.

### Thin-layer chromatography

Thin layer chromatography (TLC) was used to screen the specificity of GH29 and GH95 enzymes towards the fuco-oligosaccharides α-Fuc(1,3) GalNAc, α-Fuc(1,4)GalNAc, α-Fuc(1,3)Gal, α-Fuc(1,2)Gal, 2′FL, 3FL, Le$^a$ triose, Le$^x$ triose, Le$^b$ tetraose, LNDFH I, LNDFH II, LNFP II, LNFP V and sialyl Le$^a$ triose, while the GH33, GH181, and Amuc_0623 putative sialidase where screened on 6′SL, 3′SL and Colominic acid. Reactions (10 μL) were carried out using 2 mM of each substrate, 0.5 μM of each enzyme in 20 mM HEPES, 150 mM NaCl pH 6.8 at 37 °C for 1 h. Aliquots

of 2 μL were spotted on a silica gel 60 F254 plate (Merck, Germany) and the products were separated using a mobile phase of butanol/ethanol/Milli-Q (5:3:2, v/v/v) except for products obtained from 3′SL that were separated using a mobile phase of isopropanol/ethyl acetate/Milli-Q (3:2:1, v/v/v). The plates were dried, sprayed with 2 % 5-methylresorcinol, 80 % EtOH, and 10 % $H_2SO_4$, all v/v, and visualized by tarring at 300 °C. All enzyme activity reactions analyzed by thin-layer chromatography were performed in independent triplicates.

### Enzymatic analysis towards recombinant P-selectin glycoprotein ligand-1 (PSGL-1)/immunoglobulin mIgG2b chimeras carrying defined Lewis epitopes

To demonstrate enzymatic activity against intact mucin-type glycoproteins, PSGL-1/mIgG2b chimeras were produced and purified in glyco-engineered CHO cells as previously described[54]. In short, the extracellular portion of PSGL-1 was genetically fused with mouse immunoglobulin G2b creating the PSGL-1/mIgG2b expression plasmid, which was expressed in CHO cells together with plasmids encoding O-glycan core enzymes and combinations of fucosyl transferase genes. Thus, CHO cells were programmed to express the Lewis antigens (Le$^a$, Le$^x$, Le$^b$, or Le$^y$, Supplementary Fig. 6) on the mucin-type fusion protein. The produced PSGL-1/mIgG2b were purified from the cell culture supernatants using goat anti-mouse IgG agarose beads (Sigma−Aldrich). Each enzyme (2 μM) was incubated with beads carrying PSGL-1 glycoprotein (displaying a distinct Le antigens) in 20 mM HEPES buffer 150 mM NaCl pH 6.8 at 37 °C for 3 h in 50 μl. The beads were boiled in presence of SDS-loading buffer containing 25 mM DTT for 10 min at 95 °C. The samples were electrophoretically separated on 8% NuPAGE gels (Invitrogen, Waltham, MA, USA) and thereafter blotted onto PVDF membranes (Immobilon P membranes, 0.45 μM) according to the manufactures manual (Invitrogen). The membrane blots were blocked with phosphate-buffered saline containing 0.2% Tween-20 (v/v, PBS-T) and 3% bovine serum albumin (w/v, BSA), which was also used for the dilution of the following primary mouse antibodies, followed by washing with PBS-T twice for 5 minutes. Then each membrane was incubated with the corresponding primary antibody (see list of antibodies below, all at 1:500 dilution) for 1 h at 4 °C, washed as above, and then HRP-conjugated poly-clonal goat anti-mouse IgM (1:5000 dilution, Sigma−Aldrich) was added for 1 h at 4 °C and lastly washed with PBS-T. Bound secondary antibodies were detected by chemiluminescence using the ECL kit according to the manufacturer's instructions (GE Healthcare, Uppsala, Sweden). Finally, membranes were stripped with Restore Western Blot Stripping Buffer (Thermo Scientific) under agitation at room temperature for 20 min and re-probed with HRP-conjugated poly-clonal goat anti-mouse IgG Fc (1:5000 dilution, Sigma−Aldrich) for checking the integrity of the mouse IgG2b Fc domain of the fusion protein, then the bound antibody was visualized as above. The following primary mouse antibodies were used: IgG anti-Blood Group Lewis A (7LE) (Santa Group Biotechnology, catalog: sc-51512, 1:500), IgM anti-Blood Group Lewis B (T218) (Santa Group Biotechnology, catalog: sc-59470, 1:500), IgM anti-Lewis X antibody: CD15 (C3D-1) (Santa Cruz Biotechnology, catalog: sc-19648, 1:500), IgM anti-Blood Group Lewis Y (F3) (abcam, catalog: ab 3359). The secondary goat antibodies for Lewis A: Peroxidase goat anti-mouse IgG, F(ab′)2 (Jackson ImmunoResearch, catalog: 115-035-006, 1:5000) and for Lewis B, X, and Y: Peroxidase goat anti-mouse IgM antibody (Sigma-Aldrich A-8786, 1:5000).

### Enzymatic assay towards glycans from mucin or glycoprotein using ESI-LC MS/MS

Fucosidase activity of GH29 and GH95 enzymes was analyzed on a mixture of intact PCM, PGM, and fetuin dot-blotted on PVDF membranes or on previously release N-glycans (from human IgG) dot-blotted on membranes. Sialidase activity was assayed using released O-

glycans from PCM or PGM, or released *N*-glycans (human IgG) as well as conjugated *O*-glycans in MUC2_Mouse.

For dot-blot assays, whole mouse colonic mucin[55] or released *O*- or *N*-glycans from porcine colonic mucin or from human IgG were transferred to PVDF membranes (Immobilon P membrane, 0.45 μm) using dot blotting apparatus separately (0.1 mg per dot). Each enzyme (50 μL, 1.5 μM in 20 mM HEPES buffer with 150 mM NaCl, pH 6.8) was incubated with the substrate dots. For analyzing GH29 and GH95 fucosidases, 24 h incubations were performed, while sialidase activity was tested in 1 h (*O*-glycans from PCM and PGM, MUC2_Mouse glycoprotein) and 24 h (*O*-glycans from MUC2_Mouse glycoprotein and *N*-glycans from human IgG) incubations. Afterwards, the residual *O*-linked glycans on the dot were released by reductive amination after rinsing. The released *O*-glycans were desalted, and dried as described elsewhere[52]. The resultant glycans were purified by passage through graphitized carbon particles (Thermo Scientific) packed on top of a C18 Zip-tip (Millipore). Samples were eluted with 65% (v/v) ACN in 0.5% trifluoroacetic acid (v/v), dried, and stored at −20 °C until further enzymatic analyses.

Released glycans were resuspended in 10 μL of Milli-Q water and analyzed by liquid chromatograph-electrospray ionization tandem mass spectrometry (LC-ESI/MS) using a 10 cm × 250 μm I.D. column, packed with 5 μm porous graphitized carbon particles (Hypercarb, Thermo-Hypersil, Runcorn, UK). Glycans were eluted using a linear gradient 0–40% acetonitrile in 10 mM $NH_4HCO_3$ over 40 min at a flow rate 10 μl min$^{-1}$. The eluted *O*-glycans were detected using an LTQ mass spectrometer (Thermo Scientific, San José, CA) in negative-ion mode with an electrospray voltage of 3.5 kV, capillary voltage of −33.0 V and capillary temperature of 300 °C. Air was used as a sheath gas. Full scan (m/z 380–2000, two microscan, maximum 100 ms, target value of 30,000) was performed, followed by data-dependent MS$^2$ scans (two microscans, maximum 100 ms, target value of 10,000) with normalized collision energy of 35%, isolation window of 2.5 units, activation ρ = 0.25 and activation time 30 ms. The threshold for MS$^2$ was set to 300 counts. The data were processed using Xcalibur software (version 2.0.7, Thermo Scientific). Glycans were identified from their MS/MS spectra by manual annotation as previously described[56]. The LC-ESI/MS raw data have been deposited in Glycopost under the accession number GPST000283. The peak area (the area under the curve, AUC) of each glycan structure was calculated using the Progenesis QI software (Nonlinear Dynamics, Waters Corp., Milford, MA, USA). The AUC of each structure was normalized to the total AUC and expressed as a percentage.

## NMR spectroscopy

Substrate solutions (2.5 mM) of 6′-sialyllactose (for *Am*GH33B) or of 3′-sialyllactose (for *Am*GH181) were prepared in 50 mM MES buffer, pH 6.8 in $^2H_2O$. A 200 μL aliquot of the substrate solution was transferred into a 3 mm NMR tube and the sample was placed into an 800 MHz Bruker Avance III instrument equipped with a 5 mm TCI cryoprobe and thermally equilibrated to 310 K. The sample was tuned, matched, and shimmed in order to allow a rapid monitoring of the *Am*GH181-catalyzed conversion. The reaction was started by the addition of *Am*GH181 (1 μL, 10 μM) or *Am*GH33B (1 μL, 10 μM) into the NMR tube to a final concentration of 50 nM and mixing briefly before starting the analysis. A time series of one-dimensional $^1H$ NMR spectra was acquired to follow the reaction in real-time. The $^1H$ NMR spectra sampled 16,384 complex data points for an acquisition time of the free induction decay of 1.28 seconds. For each time point, 16 transients were summed up with an inter-scan relaxation delay of 2.0 seconds and using two dummy scans per time point, resulting in a time resolution of approximately one min. To validate the assignment of the α-Neu5Ac, the reaction was restarted and an $^1H$–$^1H$ TOCSY (2048 × 256 complex data points sampling 123 ms and 16 ms in the direct and indirect dimension, respectively) was acquired using a 10 kHz spin lock

field during a mixing time of 80 ms. The $^1H$–$^1H$ TOCSY spectrum on the reaction mixture containing intermediates of 6′-sialyllactose or 3′-sialyllactose reaction were compared with a reference standard spectrum of Neu5NAc. The NMR data were considered unambiguous when acquired in single time-series experiments. Restarted assays using $^1H$–$^1H$ TOCSY confirmed the interpretation. All NMR spectra were acquired and processed with ample zero filling using Bruker Topspin 3.5 pl7 software and were subsequently analyzed with the same software.

## Crystallization

The crystallization of *Am*GH181 (Amuc_1547) was performed by the sitting drop method using a mosquito robot (mosquito Xtal3, SPT labtech, Melbourn, United Kingdom) to mix 0.15 μL reservoir: 0.15 μL enzyme (30.5 mg mL$^{-1}$) and the plates were thereafter incubated at 18 °C. The first crystals of the enzyme with co-crystalised with ligand from Hampton Research (Aliso Viejo, CA, USA), condition 82 (0.2 M $MgCl_2 \cdot (H_2O)_6$, 0.1 M BIS-TRIS pH 5.5, 25% w/v Polyethylene glycol 3350) appeared after two weeks. After optimization, the best crystals were obtained under the same condition as above, but using a lower concentration (18% w/v) of Polyethylene glycol 3350. To evaluate if *O*-glycans from PCM could facilitate the crystallization of the enzyme with ligand, the crystallization protocol and condition as described above was used to co-crystallize *Am*GH181 with PCM. Thus, the enzyme (30.5 mg mL$^{-1}$) and PCM (4% (w/v) prepared in Milli-Q) were mixed at room temperature at a 1:1 ratio (v/v) (resulting in a final PCM concentration of 2% w/v) before the enzyme-glycan solution was mixed with the reservoir as described above and plates were incubated at 18 °C. Enzyme crystals were flash-frozen in liquid nitrogen in nylon loops using 25% ethylene glycol as cryoprotectant. Of note, the crystals from the co-crystallization appeared only after hours were much larger rhombus-shaped as compared to the crystals in the lack of added glycans.

Similarly, *Am*GH29D (25 mg mL$^{-1}$) was co-crystallized with the same glycan mixture as above and mixed with the glycan mixture (1:1 v/v). The first crystals appeared in the Molecular Dimensions structure screen 2 (Holland, OH, USA) condition 28 (0.1 M HEPES pH 7.5, 20% PEG 10,000) at 18 °C. After optimization, the best diffraction data were obtained by mixing *Am*GH29D (25 mg mL$^{-1}$) with glycans at 1:0.8 ratio (v/v) at room temperature, and reservoir condition 0.1 M HEPES pH 7.7 16% PEG 10,000, and thereafter incubation of the plate at 16 °C. Diffraction data were collected at the BioMAX beamline at the MAX IV synchrotron radiation facility (Lund, Sweden) and the P13 EMBL Beamline at the DESY (Hamburg, Germany). The data was processed with Xia2[57] using the 3dii pipeline, with XDS (version: January 31, 2020). Phasing was performed with the phaser version in included in the phenix package (v.1.19.2-4158) using an AlphaFold2 model based on the primary structure of the enzyme as a template generated using colabfold[58,59]. The structures were refined using phenix.refine[60] and manually rebuilt using Coot[61] v0.9 (Supplementary Tables 15 and 16). Structure validation was performed using MolProbity[62] version included in Phenix package v.1.19.2-4158.

## Fucosidase and sialidase activity measurements on whole cells and in culture supernatant

For localizing fucosidase and sialidase activity, *A. muciniphila* was grown in three biological triplicates anaerobically in 8 mL YCFA medium containing 0.5% (w/v) PCM for 16 h. For preparing whole cells, 12 mL culture were harvested (5000 *g*, 10 min at 4 °C) and cells were washed three times (5000 *g*, 10 min at 4 °C) with 1 mL ice-cold 10 mM sodium phosphate, 150 mM NaCl, pH = 6.5 buffer before resuspension to $OD_{600} = 0.5$ and $OD_{600} = 8$ in the same buffer. Cell suspensions, of identical volumes as above, were disintegrated by sonication (Qsonica sonicator, 5 mm probe tip, 4 × 15 s at 4 °C), and thereafter centrifuged to separate insoluble cell debris from the

clarified lysates (20,000 g for 30 min at 4 °C). The cell debris were reconstituted in the same buffer to equal volumes as the whole cell preparations to $OD_{600} = 0.5$ and $OD_{600} = 8$. To assay activity of released proteins in culture supernatants, cells were removed (20,000 g for 20 min at 4 °C) from 2 mL cultures, the supernatants were exchanged three times to the same buffer as above (Amicon Ultra 0.5 mL), 10 kDa cut off (Merck, Darmstadt, Germany) (10,000 g for 20 min at 4 °C) before adjusting to the same volumes of the assayed intact cells. Next, thin layer chromatography was used to screen for fucosidase active towards 2´FL, Le$^a$ trisaccharide, and Fuc($\alpha$1,3)GlcNAc as well as for sialidases activity towards 3´SL and 6´SL. Reactions were initiated out by mixing 10 µL of 5 mM of each substrate in the same buffer as above and 10 µL whole cell, cell debris, cell lysate, or supernatant solutions at 37 °C. Aliquots of 2 µL were spotted on a silica gel 60 F254 plate (Merck, Germany) after 1 h, 2 h, 3 h, and 4 h and the products were separated, and the plates developed as described above. The growth assays were performed in three biological triplicates and a single TLC analysis was performed for each of the three biological triplicates.

### Mucin binding assay

Binding of *A. muciniphila* GH29/GH95 fucosidases and GH33/GH181 sialidases to PGM and to Avicel (used as negative control) was assessed by a pull-down assay, followed by sodium dodecyl sulfate polyacrylamide gel electrophoresis (SDS-PAGE). In short, insoluble PGM and Avicel were washed three times (20,000 g, 5 min, 4 °C) with 1 mL standard buffer (20 mM HEPES, 150 mM NaCl, pH 6.8), before resuspension to a concentration of 1 % (w/v) in the above buffer. Next, 50 µL of the PGM or Avicel suspensions were mixed with 50 µL of fucosidases (0.1 mg mL$^{-1}$), sialidases (0.1 mg mL$^{-1}$), or bovine serum albumin (0.1 mg mL$^{-1}$) used as negative control, incubated for 20 min on 4 °C and centrifuged (20,000 g, 10 min, 4 °C). Resulting supernatants were transferred into fresh 1.5 mL reaction tubes and PGM/Avicel pellets were washed twice (20,000 g, 5 min, 4 °C) with 100 µL buffer, before resuspension in 100 µL standard buffer. Next, 100 µL protein solution was supplemented with 35 µL SDS sample buffer (NuPAGE) and the samples were boiled for 10 min, loaded (15 µL) into a gel, and analyzed using SDS-PAGE. The binding assay was performed in two independent replicates and SDS-PAGE analyses were performed once per independent replicate.

### Growth experiments, co-culture experiments and butyrate quantification

For single strain monocultures *Roseburia inulinivorans* DSM 16841, *Roseburia intestinalis* DSM 14610, *Roseburia faecis* DSM 16840, *Agathobacter rectalis* DSM 17629, *Faecalibacterium prausnitzii* DSM 17677 and *Akkermansia muciniphila* DSM 22959 were grown anaerobically at 37 °C in YCFA media using a Whitley DG259 Anaerobic Workstation (Don Whitley Scientific). Growth media were supplemented with 0.5% (w/v) carbohydrates sterilized by filtration (soluble carbohydrates, 0.45 µm filters) or autoclaving (mucins, 15 min at 121 °C) and cultures were performed in at least three independent biological replicates unless otherwise indicated. For the inhibition of *A. muciniphila* fucosidases and sialidases, culture media were supplemented with the sterile filtered (0.45 µm filters) DFJ and DANA inhibitors to a final concentration of 1 mM or 20 mM each. Bacterial growth was monitored by measuring $OD_{600}$ and for growth experiments performed in airtight sealed microtiter plates (sealing tape for 96-well plates, Thermo Scientific), a Power Wave XS microplate reader (BioTek Agilent) was used to monitor $OD_{600}$. For sialic acid and fucose quantification in culture supernatants aliquots (30 µL) were collected, mixed with 100 µL 0.9% (w/v) NaCl, before cells were removed by centrifugation (20,000 g, 4 °C 10 min). Next, supernatants were frozen at −20 °C before further analysis.

For single strain co-culture experiments of *R. inulinivorans*, *R. intestinalis*, *R. faecis*, *A. rectalis* and *F. prausnitzii* in co-culture with *A. muciniphila*, individual strains were grown in 10 mL YCFA to mid-late exponential phase ($OD_{600} = 0.6$-0.7). From these pre-cultures, equal amounts of cells ($OD_{600}$) were used to inoculate 1 mL fresh YCFA medium supplemented with 0.5% (w/v) of PCM to a start $OD_{600} = 0.01$. All cultures were performed in four independent biological replicates and growth was followed ($OD_{600}$) by sampling at 0 and 24 h. Samples (400 µL) from time 0 h and after 24 h were collected for estimating the relative bacterial abundance by qPCR and for SCFA quantification. The samples were centrifuged (20,000 g, 4 °C 10 min) and the supernatants diluted to a final concentration of 5 mM H$_2$SO$_4$, sterile filtrated (0.45 µm filters) and storage at −80 °C for SCFAs analysis. For estimating the relative bacterial abundance, the resulting cell pellets were washed once (20,000 g, 5 min at 4 °C) with 1 mL ice-cold 10 mM Tris-HCl, pH=8, resuspended in 100 µL of the same buffer and then genomic DNA was released by boiling for 15 min at 100 °C and removal of cell debris (20,000 g, 4 °C 20 min). The purified DNA was stored at −20 °C for qPCR analysis.

For mixed strain co-culture experiments of the sialic acid utilizing *R. inulinivorans* and *F. prausnitzii* or the non-sialic acid utilizing *R. intestinalis* and *A. rectalis* either alone or in co-culture with *A. muciniphila*, individual strains were grown in 10 mL YCFA to mid-late exponential phase ($OD_{600} = 0.6$-0.7). From these pre-cultures, 1 mL fresh YCFA medium supplemented with either 0,5%, 1 and 2% (w/v) of PGM or 0,5%, 1 and 2% (w/v) of BSM to start with 25% of each Clostridial strain and 50% of *A. muciniphila* to a start $OD_{600}$ of about 0.04 ($OD_{600} = 0.01$ from each Clostridium strain and $OD_{600} = 0.02$ for *A. muciniphila*). All cultures were performed in three independent biological replicates (n = 3) and growth was followed by measuring pH at 0 and 24 h. Samples (400 µL) from time 0 h and after 24 h were collected for SCFA quantification. The samples were centrifuged (20,000 g, 4 °C 10 min) and the supernatants diluted to a final concentration of 5 mM H$_2$SO$_4$, sterile filtrated (0.45 µm filters) and storage at −80 °C for SCFAs analysis.

### Butyrate quantification and estimation of relative bacterial abundance by qPCR

Butyrate in culture supernatants was quantified as previously described[47]. In short, an HPLC coupled to a refracting index detector (RID) and diode array detector (DAD) on an Agilent HP 1100 system (Agilent) was used to quantify butyrate concentrations in culture supernatants (diluted with H$_2$SO$_4$ to about 9-17% to a final H$_2$SO$_4$ concentration of 5 mM H$_2$SO$_4$) using butyric acid standards in the range 0.09-25 mM in 5 mM H$_2$SO$_4$. For analyses, 20 µL of standard or filtered (0.45 µM filter) culture supernatants samples from three biological replicates were injected on a 7.8 × 300 mm Aminex HPX-87H column (Biorad) combined with a 4.6 × 30 mm Cation H guard column (Biorad). Elution was performed with a constant flow rate of 0.6 mL min$^{-1}$ and a mobile phase of 5 mM H$_2$SO4. Standards were analyzed as above in three technical triplicates. For estimating the relative strain abundance by qPCR, SYBR green methodology (Bio-Rad) and a Rotor-Gene Q (Qiagen) PCR system were used. The extracted DNA was diluted to 1 ng µL$^{-1}$ (NanoDrop) and amplified using species-specific primers (Supplementary Table 1). The amplification mix (total 10 µL) contained 1 µL DNA, 5 µL SYBR Green Master mix (Bio-Rad), 0.22 µL of each primer (10 pmol/µL), and 3.56 µL sterile water. Amplification conditions were 1 cycle of 98 °C for 2 min, 35 cycles of 98 °C for 15 s, and 60 °C for 45 s. The melting curve analysis was performed at 60−95 °C with a ramp rate of 1 °C and 3 sec hold per step. The relative bacterial abundances in the samples were estimated based on standard curves of 0.01, 0.1, 1, and 10 ng of DNA purified from pure cultures of *A. muciniphila*, *R. intestinalis*, *R. inulinivorans*, *F. prausnitzii* and *A. rectalis*, respectively.

## Quantification and statistical analysis

For determining the statistical significance between butyrate concentrations in the different cultures, a one-Way ANOVA and Tukey post hoc test was used (OriginPro 2021). Statistical parameters, including values of $n$ and -values are reported in the figures and figure legends. The data are expressed as arithmetic means with standard deviations (SD), unless otherwise indicated. The statistical significance between growth levels reached on mucin/monosaccharides with and without the inhibitors was evaluated using an unpaired two-tailed Student's $t$-test using OriginPro 2021.

## Bioinformatics

SignalP (v.5.0), PSORTb (v.3.0.3), TMHMM (v.2.0) were used for signal peptide and transmembrane domain prediction. CAZy, dbCAN meta server, and InterPro were used under default settings to analyses size and modular organization of proteins. For phylogenetic analyses, sequences data sets were generated by identified orthologues via batch BLASTP searches using both *A. muciniphila* GH29, GH95 fucosidases or GH33/BRN repeat-like domain sialidases and those sequences that are defined as characterized in the particular GH families within the CAZy database as queries, against 7950 (meta) genomes from the human gut microbiota (retrieved from the PATRIC (v.3.6.12) database, date: November 2021, inclusion criteria: host: "human/*Homo sapiens*", isolation source: "faeces/faecal sample", genome quality "good"). Redundancy in sequence datasets was reduced using CD-HIT server under default settings and with a sequence identity cut-off = 0.95. Structure-guided protein sequence alignments were performed using PROMALS3D (http://prodata.swmed.edu/promals3d/promals3d.php) and by using structurally characterized orthologues from the particular CAZy GH families. Phylogenetic trees were constructed using the MAFFT server interfaced (https://mafft.cbrc.jp/alignment/software/) (neighbor-joining algorithm and with bootstraps performed with 1000 replicates) and afterwards visualized in iTOLv6. The prevalence of the different enzymes was analysed using 177 *A. muciniphila* good quality genomes from the same database. Identification of closest structural characterized orthologues were done using the Dali server[63]. Orthologues to *Am*GH181 were identified by a BlastP search against the nonredundant protein sequence and by using *Am*GH181 (Amuc_1547) as query sequence. Redundancy of resulting sequencing (e-value cut off: 1e−25) was reduced the using CD-HIT server under default settings and with a sequence identity cut off = 0.90 and the redundancy reduced dataset was structurally guided aligned using the PROMALS3d server (used protein structures: *Am*GH181, and structurally characterized orthologues from the GH33). *Am*GH181 orthologues were selected in resulting alignment by the presence of the conserved catalytic machinery as displayed by *Am*GH181. AlphaFold modelling was performed using ColabFold[58] v1.5.2 on its web interface (https://colab.research.google.com/github/sokrypton/ColabFold/blob/main/AlphaFold2.ipynb) under standard settings (template mode on: (Structure of *Am*GH29D); msa_mode: MMseq2, pair_mode: unpaired and paired; model_type: auto, num_recycles: 3). Sequence logos were generated using the Seq2Logo v2.0[64] web interface using standard settings (Logo type: Kullback-Leiber; Clustering method: Hobohm1; clustering threshold: 0.63 and 200 pseudo counts).

## Reporting summary

Further information on research design is available in the Nature Portfolio Reporting Summary linked to this article.

## Data availability

The LC-ESI/MS raw data have been deposited in GlycoPost depository under the accession number GPST000283. The atomic coordinates of *Am*GH29A and *Am*GH181 have been deposited in the Protein Data Bank under the PDB accessions 8AYR as well as 8AXI, 8AXS, and 8AXT,

respectively (see also Supplementary Information Tables 15 and 16). The GenPept accession IDs of the enzymes characterised in the study are enlisted in Supplementary Table 1. All the data are available from the corresponding authors upon request. The LC-ESI/MS annotated data is available in Supplementary Data file 1, and the corresponding data, where the low abundance glycan structures removed is available as Supplementary Data File 2. The data generated in this study are provided in the Supplementary Information and a Source Data file. Correspondence and requests for materials should be addressed to Maher Abou Hachem. Source data are provided with this paper.

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

## Acknowledgements

The study was funded by the Ministry of Higher Education and Scientific Research of Iraq through a Ph.D. scholarship for BS. Additional funding was from the Independent Research Fund Denmark, Natural Sciences grant 1026-00386B for MAH. The NMR spectra were recorded at the NMR Center DTU, supported by the Villum Foundation. We acknowledge MAX IV Laboratory for time on Beamline BioMax under Proposal 20200120 Research conducted at MAX IV, a Swedish national user facility, is supported by the Swedish Research Council under contract 2018-07152, the Swedish Governmental Agency for Innovation Systems under contract 2018-04969, and Formas under contract 2019-02496 and we acknowledge DESY (Hamburg, Germany), a member of the Helmholtz Association HGF, for the provision of experimental facilities. Parts of this research were carried out at P13 and we would like to thank Isabel Bento for the assistance in data collection. Beam time was allocated for proposal MX846. We acknowledge the European Synchrotron Radiation Facility for the provision of synchrotron radiation facilities and we would like to thank Daniele De Sanctis for assistance in using beamline id23-1, experimental proposal MX-2413 DOI 10.15151/ESRF-ES-708759666. The beamline was used for screening crystals towards the final crystal structures. Saromics Biostructures AB (Dr. Maria Håkansson) are thanked for the initial crystallization data. The authors would like to thank Professor Bernard Henrissat for his kind input and discussion on the defining member of GH181. We would like to thank Tina Johansen for help with the SCFA analyses.

## Author contributions

B.S., E.N.K., and M.A.H. conceptualized the research and provided funding. M.A.H. led the study and wrote the manuscript with B.S. and M.J.P. B.S. cloned and produced the enzymes for the initial TLC and LC-ESI/MS analysis. B.S. and C.J. performed the enzymatic LC-ESI/MS analyses. B.S., M.J.P., C.J., and N.G.K. analyzed and interpreted the glycomic LC-ESI/MS data. B.S., J.L., and J.H. performed and interpreted the enzyme activity on recombinant PSGL1. H.W., A.M.G., and N.J. performed and analyzed the fucosidase kinetic assays. M.J.P. and S.M. designed the NMR analysis, which was carried out by S.M., who also generated the NMR analysis figure. M.J.P. performed the microbiology growth and inhibition assay, with help from T.S.N. The structural biology work was carried out by H.S., T.S.N., and J.P.M. The final version of the figures (except for the NMR analysis) was generated by M.J.P. All authors contributed to editing the manuscript and accepted the present findings and conclusions.

## Competing interests

The authors declare no competing interests.
