## [Peer Review File · Nature Communications]

Sialidases and fucosidases of *Akkermansia muciniphila* are crucial for growth on mucin and nutrient sharing with mucus-associated gut bacteriaREVIEWER COMMENTS

Reviewer #1 (Remarks to the Author):

This manuscript describes the function and roles of the fucosidases and sialidases from *Akkermansia muciniphila* on various mucins. The study is on a topic that is very much in focus right now, and brings new knowledge that is significant for gut health and ecology, as *A. muciniphila* is a known important species in the gut microbiota. Additionally, the biochemical and structural characterization of enzymes from known families are complemented by studies of enzymes from a new CAZy family which should help its establishment in the database with a formal family number. The experiments are well-described and for the most part done appropriately. The manuscript is well-written and easy to follow and the results for the most part straightforward to interpret.

Overall, it is a very nice and interesting study, but I think there are some flaws that should be addressed, and I also have comments on stylistic choices to make the text and figures more accessible.

Major comments:

My main concern is with the co-culturing experiments. You only report OD600 here, and only from one timepoint at 24 h, as the 0 h timepoint (methods) is not really monitoring growth but just the baseline. You show *A. muciniphila* on its own growing to OD600 around 0.6 on PCM in Fig. 3o and the OD600 for the co-cultures reaching similar or somewhat lower levels. How do you know both species are growing and to which extent? Is there competition between the species? If the culture is simply reaching a maximum density, what are the proportions of each species? To make this set of experiments interpretable, I think you should first of all take more timepoints to make actual growth curves, and include species proportion determinations at each timepoint to be able to show how each species is growing. This has been done using qPCR in other works (e.g. <https://www.ncbi.nlm.nih.gov/pmc/articles/PMC4978124/>), or possibly staining and microscopy could be used. This would show whether both species are thriving and especially if *A. muciniphila* is growing as in the monoculture or if it is becoming outcompeted/disturbed. If the 24 h sample shows the maximum density in your culture volumes, a 50:50 ratio of *A. muciniphila* and the other species would not mean that the former is outcompeted if its growth rate is still the same as in the monoculture, and this is important to show to be able to talk about cross-feeding vs competition. Another way to study this would be to add your heterologously produced enzymes to cultures of the other species, though this is not as good to mimic the natural interactions, especially since the *A. muciniphila* enzymes are cell-surface located. The butyrate concentration measurements are a very nice complementary analysis but does not necessarily directly correlate with abundant growth. Lastly, the much higher OD600 for *A. muciniphila* growing on PCM in panel a vs panel o would be good to comment (I assume different PCM batches were used for these experiments?)

Mucin oligos: the experiments that are reported in Fig. 1 and similar are very nice, but I think it is a bit complicated to interpret some of the results. The disappearance of a glycan structure is very clear and nicely shows the action of an enzyme, but in other instances you have an increase of oligosaccharides, which then would come from some precursor on which the enzyme acts. I think to make the figure and data easier to present, you should describe from where these might come, at least when you have increase of an oligosaccharide. This can be done in the text, or maybe you could include a symbol in your panels for each glycan structure, for example in white, or dashed, to indicate the presumed bond that has been cleaved to obtain this structure.

Structures: first, I would like to congratulate the authors on the very nice structural work and interesting apo and ligand structures. I think though that some additional figures/analyses would improve the manuscript. First of all, why do you only show the catalytic domain alphafold (AF2) model for GH29C (paragraph line 220)? It should have the same architecture as GH29D, so why not show the AF2 model of the whole enzyme? It would be nice to see if it has the same cobra pose or not – either way it would be of interest. Also, I think the AF2 model should also be shown as a cartoon to be able to appreciate the different loop lengths you describe. I personally think it would also be interesting to see an AF2 model of GH29B as it seems to contain quite large regions with no annotation – are these predicted to have secondary structure elements or are they disordered?

Mechanism of GHxxx: it is great that you have a structure of a GHxxx enzyme and you discuss its active site and mechanism and relation to GH33 at some length. I think this discussion and pointing out presumed catalytic residues warrants some investigation. It should be straightforward to make simple Ala variants of the predicted catalytic residues to verify that they are indeed involved in the catalysis, especially since you point out a quite striking substitution of the catalytic Tyr nucleophile in GH33 enzymes. A 'simple' activity difference measurement based on specific activity could suffice, in my opinion, to show for example a 1000-fold reduction in activity or similar.

Minor:

Your family abbreviations, e.g. GH33 stands for glycoside hydrolase family 33. It does not stand for glycoside hydrolase family 33 enzyme. Therefore, you cannot write GH33s or GH95s but have to include enzyme/protein/member or similar as in GH33 enzymes (otherwise you're basically saying that you used several GH33 families which makes no sense). Or, re-define what the abbreviation stands for. There are several instances throughout the manuscript, but these should be easy to address. The same goes for CBM families, as in "joined to a CBM32" → "joined to a CBM32 protein".

From your descriptions of the cloning and looking at your primers, you cloned the whole genes apart from the signal peptides. I think you should spell this out more clearly by adding e.g. "the full-length genes were cloned" to the text. It is nice to show that you were able to use the whole enzyme and did not have to focus on "only" the catalytic domains.

I think you should use s-1 instead of min-1 for the kinetic parameters. That is by far the more common way to report kcat values.

Supplementary tables 4 and 6: please write the incubation time also in the table description.

Supplementary figure 2: remove spaces between CBM and the family number (e.g. CBM 32 should be CBM32).

Line 127: remove s from fucosidases.

Line 256: where is this D324? It is not in the sequence logo or structure figure. Should be D345?

Line 264: same as above, why did W298 change to W277??

Line 267: sugar tong? Surely not tang, at least.

Line 305: also cell lysis could be a reason for release of enzymes and is quite common.

Line 357: change remarkably to interestingly/curiously or similar. I don't think remarkable is the right word, as it can read as wondrous or amazingly which doesn't really fit.

Reviewer #2 (Remarks to the Author):

The manuscript describes the characterization of the activity and structure of the sialidases and fucosidases of the mucin-specialist *Akkermansia muciniphila*. The work is well-structured and findings are well founded by experiments, however we do have some concerns.

Major revisions

- Albeit that the manuscript is well-structured, the whole manuscript needs attention regarding writing style and language. There are several sentences that are incomplete or do not flow well. Also, as will be clear from the comments below, several parts of the manuscript suffer from unclear and overstatements. These should be addressed in order to deliver a well-balanced and discussed work.

- The abstract needs to be rewritten, there are several unclarities and overstatement in the current version. Some examples. Line 24: *A. muciniphila* is associated to the integrity of the mucus layer. Although this bacterium is a mucin-degrading specialist and is an important microbiota member at the interface between mucin and the microbiota, claiming it is associated to the integrity of the mucus layer itself, is an overstatement. This should be rephrased. Line 31: sialidases and fucosidases confer mucin-binding, this is an overstatement. Albeit that these CAZymes have mucin-binding activity, it remains elusive if they are the ones conferring the mucin-binding activity of *A. muciniphila*, especially since the claim that the CAZymes are cell-attached is poor (see below). Line 32: fucose is not a sialic acid, rephrase. Line 35, better than to point towards nutrient sharing, best is to mention the role of *A. muciniphila* at the start of several trophic/cross-feeding chains.

- The introduction of the paper has several issues.

Line 38: instead of GM, it is better to use standard abbreviations used in the field, like MB, or none at all. Line 39: the statement on fiber intake is not supported by literature, Albeit that fiber intake is associated to the microbiome, it is not the only one and a fiber-poor diet per se does not lead to breaches in the gut barrier. This is a faulty overinterpretation of the literature. Also, this statement does not fit in the flow of the introduction.

Muc2 and the other mucin proteins are mostly written in capital letters, MUC2.

Line 45: glycan chains or molecules, instead of merely glycans

Line 46: this sentence needs to be rephrased as in its current form it reflects a wrong statement.

Line 48: mention the mucus layer is largely sterile

Line 49: rephrase this sentence for clarity

Line 60: notably should be deleted from this sentence

Line 64: we disagree with the notion that this is largely uncharted territory as various papers and research endeavors have worked on the characterization of the glycan degrading apparatus of *A. muciniphila*. Stating this is an uncharted territory is a grove overstatement and neglect of the prior work done in this field.

In globo, this introduction needs to be reworked significantly prior to publication of this work.

- Although it makes sense to indicate the families of the CAZymes that were further characterized in this work, it is more common to refer to their locus tag in *A. muciniphila*, and mentioning or referring back to these locus tags, would enhance the clarity of the work.

- Figure 1 and 2: A better explanation of the "relative abundance" shown in these figures is needed for clarity and dissection of the results. The current explanations are not clear enough.

- Line 168: we suggest to rephrase this sentence, as we agree that Amuc_1547 might be the defining member of a new GHxx family, it is better to state this more prudently, given that before a new family is accepted or added to the CAZy database, a thorough revision of all evidence is warranted. Therefore, we would like to see this statement revised

- Line 305: How can you be sure these enzymes are cell-attached and released by proteolysis and not secreted? We do not see where the notion of enzyme release due to proteolysis comes from. Please revise. This also applies to line 384.

- The discussion section of the paper is very limited and should be elaborated more.

Minor revisions

- Please doublecheck the correct writing of the HMO denominations, as there are several errors throughout the paper, e.g. 6sL.

- Line 393: ...was commensurate... can you rephrase this for clarity?

Reviewer #3 (Remarks to the Author):

In the manuscript by Shuoker et al., the authors express and characterize a series of fucosidases and sialidases from *Akkermansia muciniphilia*, a known mucin degrader. They then go on to crystallize several of these enzymes, thus revealing structural reasoning behind the glycosidases' specificities.

Overall, this is an incredibly thorough manuscript which is important and adds a substantial amount of information to what is currently known. It is well-written, the story is straightforward, and I largely support publication of the article with relatively minor edits.

- Please specify the GH29/95 function in the first paragraph
- Why wasn't Muc2 used for the fucosidase assays?
- In Supplemental Fig 5, why aren't all of the enzymes depicted in each graph?
- The glycan digestion assays are not very well described – while I found this information in the methods, it would be useful to have 2-3 sentences describing how relative abundances are calculated, that this is performed using mass spec, etc. Also, how are relative abundances >100%? Is this compared to control?
- In general, the figure captions do not have enough experimental detail. It would be useful to assure that each figure caption (main body and supplemental) describe how the experiments were performed/data visualized.
- Similarly, I am left confused as to how glycans are detected when digested off of intact proteins. Can the authors add a discussion of this to the Results section?
- In Supplemental Fig 6, where is the control? Ladder?
- Supplemental Table 4 – how are the percentages of fucose higher than in the fucosidase treated samples?
- Might not be possible, but Supp Fig 8e is really tough to see, would it be possible to darken or up the contrast on the image?
- In Supp Fig 17, please describe what the different colors and boxes indicate. Also, what is indicative of enrichment? And how was the enrichment performed?
- In Supp Fig 18, how were the glycans isolated? Is this Western blot or TLC? Why was there no activity in the cell lysate, as if the enzymes are membrane tethered I would still expect activity?
- The sentence "fucosidase and sialidase was critical for mucin destruction" is not supported by Fig 4A, B, or the data in figure 4. Please remove or elaborate on how fig 4 shows this.

Detailed response to the referee report

We would like to thank all three referees for the great effort and valuable feedback on our manuscript. Please find our detailed responses below and we are confident that we have addressed all the concerns and hopefully the revised manuscript will be markedly improved due to your insightful review. Please note that we made a variety of minor cosmetic corrections/clarifications of the revised manuscript, beyond the specific requests you kindly raised, which we hope will improve clarity.

Reviewer #1 (Remarks to the Author):

This manuscript describes the function and roles of the fucosidases and sialidases from *Akkermansia muciniphila* on various mucins. The study is on a topic that is very much in focus right now, and brings new knowledge that is significant for gut health and ecology, as *A. muciniphila* is a known important species in the gut microbiota. Additionally, the biochemical and structural characterization of enzymes from known families are complemented by studies of enzymes from a new CAZy family which should help its establishment in the database with a formal family number. The experiments are well-described and for the most part done appropriately. The manuscript is well-written and easy to follow and the results for the most part straightforward to interpret. Overall, it is a very nice and interesting study, but I think there are some flaws that should be addressed, and I also have comments on stylistic choices to make the text and figures more accessible.

Major comments:

1. My main concern is with the co-culturing experiments. You only report OD600 here, and only from one timepoint at 24 h, as the 0 h time point (methods) is not really monitoring growth but just the baseline. You show *A. muciniphila* on its own growing to OD600 around 0.6 on PCM in Fig. 3o and the OD600 for the co-cultures reaching similar or somewhat lower levels. How do you know both species are growing and to which extent? Is there competition between the species? If the culture is simply reaching a maximum density, what are the proportions of each species? To make this set of experiments interpretable, I think you should first of all take more timepoints to make actual growth curves, and include species proportion determinations at each timepoint to be able to show how each species is growing. This has been done using qPCR in other works (e.g. <https://www.ncbi.nlm.nih.gov/pmc/articles/PMC4978124/>), or possibly staining and microscopy could be used. This would show whether both species are thriving and especially if *A. muciniphila* is growing as in the monoculture or if it is becoming outcompeted/disturbed. If the 24 h sample shows the maximum density in your culture volumes, a 50:50 ratio of *A. muciniphila* and the other species would not mean that the former is outcompeted if its growth rate is still the same as in the monoculture, and this is important to show to be able to talk about cross-feeding vs competition. Another way to study this would be to add your heterologously produced enzymes to cultures of the other species, though this is not as good to mimic the natural interactions, especially since the *A. muciniphila* enzymes are cell-surface located. The butyrate concentration measurements are a very nice complementary analysis but does not necessarily directly correlate with abundant growth.

Response: Thank you for the insightful questions. Our colonic mucin batch was depleted and it was not feasible within to prepare new substrate within a reasonable timeframe for the re-submission. Instead we set up a qPCR assay to measure the proportion of the coculture start (t=0 h) and after 24 h. These data showed a reduction of the relative abundance of the Clostridia strains drops from about 50% at start to less than 10% of the culture after 24 hours. Two of the strains that were unable to grow on sialic acid were not detectable in the qPCR assay, while the third was as low as about 3%. These data do not support a competition with *A. muciniphila*, and they are consistent with our data in Fig. 4, and on the low proportion of sialic acid relative to the total mucin concentration. To evaluate the correlation between sialic acid and butyrate production, we performed a different experiment

using PGM (sialic acid content 0.5-1.5% w/w according to manufacturer) and bovine submaxillary mucin (BSM; sialic acid 9-24% w/w according to manufacturer). These experiments showed higher butyrate concentrations on the more densely sialylated BSM. Collectively this shows that the generation of butyrate is mainly due to crossfeeding, mainly one sialic acid, and not due to competition on mucin between *A. muciniphila* and the Firmicutes. These data are added as SI Fig. 19b (qPCR) and Fig. 20 (PGM/BSM comparison)

2. Lastly, the much higher OD₆₀₀ for *A. muciniphila* growing on PCM in panel a vs panel o would be good to comment (I assume different PCM batches were used for these experiments?)

Response: The used mucin substrate is the same. In the graphs in panels “a,b”, the OD₆₀₀ is measured in cuvettes (l=1cm), whereas the data in the other panels were measured in 96-well plates, with a shorter light path, so growth levels are very similar. This is clarified in the legend

3. Mucin oligos: the experiments that are reported in Fig. 1 and similar are very nice, but I think it is a bit complicated to interpret some of the results. The disappearance of a glycan structure is very clear and nicely shows the action of an enzyme, but in other instances you have an increase of oligosaccharides, which then would come from some precursor on which the enzyme acts. I think to make the figure and data easier to present, you should describe from where these might come, at least when you have increase of an oligosaccharide. This can be done in the text, or maybe you could include a symbol in your panels for each glycan structure, for example in white, or dashed, to indicate the presumed bond that has been cleaved to obtain this structure.

Response: There are minute errors associated with dispensing 1 μ L of mucins, which are surface-active materials, onto PVDF membranes for enzymatic incubations. Consequently, differences between controls and enzymatic incubations may be amplified for quantitatively minor glycan structures. For example, a minor glycan would be at or below detection in the control, but observed in the reaction, which may result in artefactual differences in relative abundance, especially if the structure persists in the enzyme incubated pool, due to lack of enzyme activity. This noise, however, is not expected to affect the overall enzyme activity profiles due to the typically large number of glycan structures analysed with many of them carrying the same epitope. We have validated this by removing glycan structures that have less than 0.5% relative abundance, which essentially suppresses this artefact. This data will be uploaded as an extended data file with the revised manuscript. However, we feel more comfortable not manipulating the data in the manuscript, e.g. by dumping some of the data. Instead, we have displayed the summed data for specific epitopes to be transparent, without giving disproportionate weight to potential artefacts. This way to present the data shows that the difference between controls and incubations with inactive enzymes are minor and that the overall activity profiles reported are robust.

For certain glycans, the increase simply depends on the enzymatic activity on a precursor to the glycan structure, e.g. the activity of GH95/GH29 on Le^{b/y} results in an increase in the mono-fucosylated structures. Both these explanations have been incorporated in the Figure/Table legends, where appropriate and references to the extended data file “Extended data file 2: Changes in abundance of glycan structures that have a relative abundance >0.5%” will also be given

4. Structures: first, I would like to congratulate the authors on the very nice structural work and interesting apo and ligand structures.

I think though that some additional figures/analyses would improve the manuscript. First of all, why do you only show the catalytic domain alphafold (AF2) model for GH29C (paragraph line 220)? It should have the same architecture as GH29D, so why not show the AF2 model of the whole enzyme? It would be nice to see if it has the same cobra pose or not – either way it would be of interest. Also, I think the AF2 model should also be shown as a cartoon to be able to appreciate the different loop lengths you describe. I personally think it would also be interesting to see an AF2 model of GH29B as

it seems to contain quite large regions with no annotation – are these predicted to have secondary structure elements or are they disordered?

Response: Thank you for the positive response and great point. The AlphaFold Model of *AmGH29C* had a more closed conformation that resembles a “Cobra bite pose”, due to a rigid body movement of the linker-CBM32 domain, positioning the CBM32 at the side of the active site. Analysing the sequences of the linkers, we observed that the loop joining the catalytic domain to the linker domain differ, especially in regard to a potential three peptide flexible hinge in *AmGH29C* (AGA), which may confer higher flexibility than a “VIL” hydrophobic cluster that is likely to contribute to the rigidity of the *AmGH29D*. Nonetheless, the Cobra-strike pose of *AmGH29D* appears conformationally feasible also for *AmGH29C* based on modelling of the latter enzyme in coot using the crystals structure of *AmGH29D* as a template. The CBM32 of both enzymes overlaid well and the linker region was modelled with the 2Fo-Fc map from the *AmGH29D* structure as reference. These observations have been incorporated in the manuscript in SI Fig. 11 and they are described in the results. For *AmGH29B*, indeed there three unassigned domains in the available AF model (Accession: AF-B2ULU6-F1), two N-terminal and one C-terminal to the catalytic domain. Since we have no data on the specificity of this enzyme, we are reluctant to include the model in the manuscript as it will not add value

5. Mechanism of GHxxx: it is great that you have a structure of a GHxxx enzyme and you discuss its active site and mechanism and relation to GH33 at some length. I think this discussion and pointing out presumed catalytic residues warrants some investigation. It should be straightforward to make simple Ala variants of the predicted catalytic residues to verify that they are indeed involved in the catalysis, especially since you point out a quite striking substitution of the catalytic Tyr nucleophile in GH33 enzymes. A ‘simple’ activity difference measurement based on specific activity could suffice, in my opinion, to show for example a 1000-fold reduction in activity or similar.

Response: Thanks for the insightful suggestion. Since this enzyme will be the defining member of a new family, we described the structure in some detail w.r.t .differences to GH33 and conservation at the active site. We also provided NMR evidence for the inverting mechanism that is consistent with the presence of a water molecule instead of the O atom of the Tyr sidechain, deduced from the structural analysis. Compelling assignment of the catalytic residues and their roles may require a suite of mutants, including single, double and triple mutants, since the Q/H residues are co-conserved and since there are additional changes in the active site compared to GH33 members. We agree that single alanine scanning would certainly bring insight into the impact of specific residues on catalysis. This however, may not be enough to unequivocally establish the catalytic roles, as several residues may contribute to the positioning and the activation of the catalytic water in the inverting mechanism, e.g. Q350, R234, H349, E218. We think that establishing the role of individual amino acids in the catalytic chassis deserves a dedicated study that also includes computational MD-MM/QM evidence. We are planning such a study, but we feel it is beyond the scope of this work centred on the biological roles and the determinants of specificity of the analysed enzymes. We have merely limited the text to the description of the conservation of the residues at the catalytic site and removed speculations on their roles in the mechanism to maintain rigor

Minor:

Your family abbreviations, e.g. GH33 stands for glycoside hydrolase family 33. It does not stand for glycoside hydrolase family 33 enzyme. Therefore, you cannot write GH33s or GH95s but have to include enzyme/protein/member or similar as in GH33 enzymes (otherwise you’re basically saying that you used several GH33 families which makes no sense). Or, re-define what the abbreviation stands for. There are several instances throughout the manuscript, but these should be easy to address. The same goes for CBM families, as in “joined to a CBM32” → “joined to a CBM32 protein”.

Response: This is done

From your descriptions of the cloning and looking at your primers, you cloned the whole genes apart from the signal peptides. I think you should spell this out more clearly by adding e.g. “the full-length genes were cloned” to the text. It is nice to show that you were able to use the whole enzyme and did not have to focus on “only” the catalytic domains.

Response: This is done “We produced and purified all six full-length enzymes.”

I think you should use s-1 instead of min-1 for the kinetic parameters. That is by far the more common way to report kcat values.

Response: This is done and we corrected some issues with error propagation in all SI tables reporting kcat or normalised activities

Supplementary tables 4 and 6: please write the incubation time also in the table description.

Response: This is added to the Tables

Supplementary figure 2: remove spaces between CBM and the family number (e.g. CBM 32 should be CBM32).

Response: The spaces between the CBM and the family number have been removed

Line 127: remove s from fucosidases.

Response: This is corrected

Line 256: where is this D324? It is not in the sequence logo or structure figure. Should be D345?

Response: The sequence numbering has been corrected

Line 264: same as above, why did W298 change to W277??

Response: Thanks for catching this error. The sequence numbering has been corrected

Line 267: sugar tong? Surely not tang, at least.

Response: Thanks, indeed tong!

Line 305: also cell lysis could be a reason for release of enzymes and is quite common.

Response: We agree, but we have omitted this phrasing altogether for better rigour.

Line 357: change remarkably to interestingly/curiously or similar. I don't think remarkable is the right word, as it can read as wondrous or amazingly which doesn't really fit.

Response: Changed to “curiously”

Reviewer #2 (Remarks to the Author):

The manuscript describes the characterization of the activity and structure of the sialidases and fucosidases of the mucin-specialist Akkermansia muciniphila. The work is well-structured and findings are well founded by experiments, however we do have some concerns.

Major revisions

1. Albeit that the manuscript is well-structured, the whole manuscript needs attention regarding writing style and language. There are several sentences that are incomplete or do not flow well. Also, as will be clear from the comments below, several parts of the manuscript suffer from unclear and overstatements. These should be addressed in order to deliver a well-balanced and discussed work.

Response: Thank you for the constructive feedback

2. The abstract needs to be rewritten, there are several unclarities and overstatement in the current version. Some examples. Line 24: *A. muciniphila* is associated to the integrity of the mucus layer. Although this bacterium is a mucin-degrading specialist and is an important microbiota member at the interface between mucin and the microbiota, claiming it is associated to the integrity of the mucus layer itself, is an overstatement. This should be rephrased.

Response: We thank the reviewer for the comment. Our statement is based on 1) Increase in the number of mucin-secreting goblet cells in *Akkermansia*-administered high fat diet-fed mice (see Shin NR, et al. *Gut* 63, 727-735 (2014) and 2) *A. muciniphila* restored the thickness of the inner mucus layer in fat diet-fed mice (Everard A, et al *Proc Natl Acad Sci U S A* 110, 9066-9071 (2013). In case of dysbiosis, levels of *A. muciniphila* and other non-mucolytic clostridial Firmicutes are decreased, which is a possible marker for the erosion of the mucus layer and the exposure of the underlying epithelial for inflammatory cues. As this evidence is indirect, we have rephrased the sentence to: "The mucolytic human gut microbiota specialist *Akkermansia muciniphila*, which is proposed to boost mucin-secretion by the host, is a key player in mucus turnover."

3. Line 31: sialidases and fucosidases confer mucin-binding, this is an overstatement. Albeit that these CAZymes have mucin-binding activity, it remains elusive if they are the ones conferring the mucin-binding activity of *A. muciniphila*, especially since the claim that the CAZymes are cell-attached is poor (see below). Line 32: fucose is not a sialic acid, rephrase.

Response: We swapped "conferred" to "displayed", so the sentence reads: "Cell attached sialidases and fucosidases displayed mucin-binding and their inhibition abolished growth of *A. muciniphila* on mucin." We disagree, however, that the claim on cell-attachment is poor as we argue below

4. Line 35, better than to point towards nutrient sharing, best is to mention the role of *A. muciniphila* at the start of several trophic/cross-feeding chains.

Response: We agree that the trophic interactions of *A. muciniphila* are likely to be more complex than we demonstrate here for the selected strains. However, we chose to focus on the evidenced observed syntrophy based on our evidence. As the reviewer would see we have further supported the syntrophy with 1) qPCR relative abundance estimates to show that no competition is occurring and 2) comparing two additional mucins with different sialic acid substitution density (see answer to Reviewer 1, point 1)

5. The introduction of the paper has several issues. Line 38: instead of GM, it is better to use standard abbreviations used in the field, like MB, or none at all.

Response: We do not share the impression that a standard abbreviation is being used generically, but we agree to omit the abbreviation for clarity

6. Line 39: the statement on fiber intake is not supported by literature, Albeit that fiber intake is associated to the microbiome, it is not the only one and a fiber-poor diet per se does not lead to breaches in the gut barrier. This is a faulty overinterpretation of the literature. Also, this statement does not fit in the flow of the introduction.

Response: We respectfully agree that fibre-microbiota interplay is not the only factor. We have not claimed diet is the only factor, but we claimed it is an important factor, based on the erosion of the mucus layer as a consequence of low fibre intake (reference: Desai MS, *et al.* A dietary fiber-deprived gut microbiota degrades the colonic mucus barrier and enhances pathogen susceptibility. *Cell* 167, 1339-1353.e1321, 2016). This likely happens due to the fact that certain bacterial groups switch their metabolism to mucin in the absence of dietary fibre (see Sonnenburg JL, *et al.* Glycan foraging *in vivo* by an intestine-adapted bacterial symbiont. *Science* 307, 1955-1959 (2005). The destruction of the mucus is a key precursor to the breach of the barrier function due to subsequent exposure to molecular pro-inflammatory cues. While our statement maybe incomplete, it is not clear to us what is the “faulty interpretation” that the kind reviewer suggests. However, as we agree that the sentence is not essential in this position, and we have deleted it

7. Muc2 and the other mucin proteins are mostly written in capital letters, MUC2.

Response: This is changed

8. Line 45: glycan chains or molecules, instead of merely glycans

Response: Done

9. Line 46: this sentence needs to be rephrased as in its current form it reflects a wrong statement.

Response: The statement is rephrased to “Similarly to other mucins, MUC2, is an O-glycoprotein that is secreted by intestinal goblet cells and consists of up to 80% (w/w) glycan chains⁶ that exhibit large structural diversity (>100 structures reported)⁷”

10. Line 48: mention the mucus layer is largely sterile

Response: This claim that the mucus layer is sterile is not accurate, only the inner mucus layer is sterile in healthy subjects, so we rephrased to “thereby creating variable adhesion and nutritional niches for the microbiota on the outer mucus surface, while the inner mucus layer in the colon is sterile”. We also moved this sentence to the new paragraph describing the microbial interactions with mucin

11. Line 49: rephrase this sentence for clarity

Response: The sentence is rephrased see the answer above. We also clarified the sentence on the capping gradients

12. Line 60: notably should be deleted from this sentence

Response: Done

13. Line 64: we disagree with the notion that this is largely uncharted territory as various papers and research endeavors have worked on the characterization of the glycan degrading apparatus of *A. muciniphila*. Stating this is an uncharted territory is a grove overstatement and neglect of the prior work done in this field.

Response: We are aware of several publications addressing, genomics, proteomics, transcriptomics, exo-glycosidase structures and basic biochemical characterisation of enzymes on simple model substrates, (e.g. work by Prof Voglmeir *et al.*). To date, a single study, by the Crouch/Bolam team, has characterized GH16 endo-glycosidases from *A. muciniphila* on mucin. However, we are not aware of a single study describing the mucin O-glycan preferences of *A. muciniphila* exo-glycosidases, which is precisely what we are claiming, but we modified the phrasing for clarity. In case we have overlooked any published work that challenges our claim, we would be very grateful if the kind reviewer points us to such work, in which case we will be happy to acknowledge our error and rectify the statement to give due credit to the scientists we have potentially overlooked. We have added a sentence to clarify and cited two references on structure/biochemistry but not on mucin complex O-glycans or intact mucins. The text in the revised manuscript reads “Several *A. muciniphila* CAZymes have been

structurally and biochemically characterized using simple oligosaccharide substrates^{19,20}. By contrast, the specificities of *A. muciniphila* exoglycosidases, and notably the sialic acid and fucose decapping apparatus, towards mucin *O*-glycans remain unexplored.”

14. In globo, this introduction needs to be reworked significantly prior to publication of this work.

Response: We have introduced several modifications to add clarity and avoid ambiguous interpretations

15. Although it makes sense to indicate the families of the CAZymes that were further characterized in this work, it is more common to refer to their locus tag in *A. muciniphila*, and mentioning or referring back to these locus tags, would enhance the clarity of the work.

Response: Locus tags bear no functional information and they are subject to changes sometimes if the genomes are revised, so we reluctant to abandon formal nomenclature. The translation between locus tags, enzyme CAZy names and genbank accessions (see SI Table 1 and Fig. S1) provides a clear link to other works that have used accessions or locus tags

16. Figure 1 and 2: A better explanation of the “relative abundance” shown in these figures is needed for clarity and dissection of the results. The current explanations are not clear enough.

Response: We agree and this has been added to the legend of the figure “The relative abundances were calculated by integration of the ESI-LC MS/MS ion chromatogram peak (area under the curve, AUC) of each glycan and normalizing it to the total (expressed in %).”

17. Line 168: we suggest to rephrase this sentence, as we agree that Amuc_1547 might be the defining member of a new GHxx family, it is better to state this more prudently, given that before a new family is accepted or added to the CAZy database, a thorough revision of all evidence is warranted. Therefore, we would like to see this statement revised

Response: The evidence for the establishment of the new family has already been presented and discussed with the CAZy curators, who accepted this. The new family has been already prepared by the CAZy team and will be posted as soon as the publication is accepted. We have rephrased the sentence to make it more clear: “Based on the structural divergence, functional, and mechanistic differences to GH33 sialidases as described below, the enzyme encoded by Amuc_1547 (*AmGHxxx*) is proposed as the defining member of the CAZy family GHxxx.”

18. Line 305: How can you be sure these enzymes are cell-attached and released by proteolysis and not secreted? We do not see where the notion of enzyme release due to proteolysis comes from. Please revise. This also applies to line 384.

Response: The earliest and highest fucosidase activity was detected in the intact cell fraction (see SI Fig. 18a-b, g), which strongly supports cell attachment. Only minor or no fucosidase activity was observed in the supernatant at low OD₆₀₀ value, which we deem as a likely artefact due to release of enzymes to the supernatant during the long assay incubation carried out aerobically. We speculated that release at high cell density/prolonged incubation could be due to proteolysis (in the linker region between the enzyme and point of anchoring at the cell surface) or cell-lysis, but we have no evidence to support the release mechanism, so we deleted this speculation from the text. Our conclusion is that early detection of clear fucosidase activity with the intact cell fraction and not in the supernatant is consistent with the cell-attachment of the fucosidases. The sialidase activities are distributed between the cell fraction and the culture supernatant from the start of the incubation. Again, these data suggest that at least one of the GH33 enzyme is cell attached and at least one is secreted. We carried out a new growth with 3'SL and 6'SL as a control, which showed similar results. Therefore, we have

included this confirmatory experiment in the SI Fig. 18. This information has also been indicated in an additional model Fig. 5

19. The discussion section of the paper is very limited and should be elaborated more.
Response: The discussion has been expanded to address the potential rationale for the evolution of an enzyme panel with various degrees of promiscuity. This includes a hypothesis that the sialidases are de-shielding the recognition sites for *A. muciniphila* glycopeptidases, based on recent findings that these glycopeptidase specifically target the sialylated T and Tn antigens. We also cited recent work that presents a model for mucin breakdown by *Akkermansia muciniphila* that involves internalised mucin oligomers, consistent with our data and model of cell attached decapping enzymes. This has been visualised in a model main body Fig. 5 that summarises our finding and recent literature on glycopeptidases and the proposed mechanism for mucin utilisation

Minor revisions

- Please doublecheck the correct writing of the HMO denominations, as there are several errors throughout the paper, e.g. 6sL.

Response: Thank you for the comment, the use of lower case “s” is not an error, but was chosen to avoid mixed-up between sialyl and sulphatyl designations, with both caps being commonly abbreviated with a capital “S”. We chose to change the sialyl to capital and the sulphatyl to lower case to make this less confusing for the very established sialo-oligo designation within the glyco-community. We also corrected the missing “” for the SL and FL substrates

- Line 393: ...was commensurate... can you rephrase this for clarity?

Response: This is rephrased

Reviewer #3 (Remarks to the Author):

In the manuscript by Shuoker et al., the authors express and characterize a series of fucosidases and sialidases from *Akkermansia muciniphila*, a known mucin degrader. They then go on to crystallize several of these enzymes, thus revealing structural reasoning behind the glycosidases’ specificities. Overall, this is an incredibly thorough manuscript which is important and adds a substantial amount of information to what is currently known. It is well-written, the story is straightforward, and I largely support publication of the article with relatively minor edits.

• Please specify the GH29/95 function in the first paragraph

Response: A brief explanation has been added to the first paragraph in the results

• Why wasn’t Muc2 used for the fucosidase assays?

Response: Mouse Muc2 has dominant terminal H type (α 1,2Fuc) and Sda [NeuAc α 2-3(GalNAc β 1-4)Gal] epitope on O-glycans (Arike, L., 2017, *Glycobiology*, 27:318; Holmen Larsson, J. M., 2013, *American Journal of Physiology-Gastrointestinal and Liver Physiology*, 305:G357). The level of other types of fucosylations (e.g., Lewis and blood group) except core 1,6-fucosylation on N-glycans is very low in mouse MUC2. Thus, this substrate was only used for sialidase enzymatic assays

- In Supplemental Fig 5, why aren't all of the enzymes depicted in each graph?

Response: The GH95 regio-selectivity is exclusive to α 1,2-linkages, therefore we just did not include the panels related to those enzymes on motifs with non-substrates as that does not add information. This is clarified in the legend

- The glycan digestion assays are not very well described – while I found this information in the methods, it would be useful to have 2-3 sentences describing how relative abundances are calculated, that this is performed using mass spec, etc. Also, how are relative abundances >100%? Is this compared to control?

Response The explanations have been added to both main body and SI Fig. legends. The more than 100% is either an artefact due to the noise in spotting the mucin, in certain case, whereas the release of precursor to yield the increased epitopes are the cause in other cases (see explanation for Referee 1, point 3)

- In general, the figure captions do not have enough experimental detail. It would be useful to assure that each figure caption (main body and supplemental) describe how the experiments were performed/data visualized.

Response: The figure legends have been clarified. With respect to the explanation of glycan > 100%, are mainly due to artefacts of mucin spotting (see answer to review 1, major point 1, who asked the same question). Another reason, is that the enzyme activity from a precursor generates additional amounts of a specific glycan. The legends in the revised version are modified to clarify either case

- Similarly, I am left confused as to how glycans are detected when digested off of intact proteins. Can the authors add a discussion of this to the Results section?

Response: The mucins are blotted on membranes and these are either incubated with buffer or with specific enzymes. Following this, the glycans are released by reductive β -elimination and analysed in the MS. This is described in the materials and methods, but it is mentioned briefly in the legends of the revised version to clarify

- In Supplemental Fig 6, where is the control? Ladder?

Response: Since the enzyme activities are known from the TLC analysis, these data are merely confirming the regio-selectivity which cannot be unambiguously assigned in the MS analysis. The activity or lack of activity of multiple enzymes serve also as positive and negative controls for each experiments. This has been clarified in the legend. The experiments monitor major changes or depletion of the specific signals that arise from the previously well-characterised cells that express single well-defined Le epitopes. The change in size is not relevant to the analysis, and that is why there is no ladder

- Supplemental Table 4 – how are the percentages of fucose higher than in the fucosidase treated samples?

Response: Sorry, we are not sure we understand the question. The percentages are of *O*-glycans and not for fucose. If what is meant refers to column 2 (non-fucosylated), then this is logical as the enzymatic activities increase the proportion of non-fucosylated structures, except for the inactive enzymes *AmGH29A* and *AmGH29B* which are inactive and the changes reflect the noise of the experiment

- Might not be possible, but Supp Fig 8e is really tough to see, would it be possible to darken or up the contrast on the image?

Response: The contrast has been increased in the revised version to add clarity

- In Supp Fig 17, please describe what the different colors and boxes indicate. Also, what is indicative of enrichment? And how was the enrichment performed?

Response: The colouring schema of the enzymes and the concept of the binding are briefly described in the legend in the revised version. For clarity, we additionally simplified the colouring scheme by removing the colour of the substrates. The fact that the enzymes are depleted from the supernatant and more heavily observed in the mucin insoluble fraction justify the claim of the mucin binding activity

- In Supp Fig 18, how were the glycans isolated? Is this Western blot or TLC? Why was there no activity in the cell lysate, as if the enzymes are membrane tethered I would still expect activity?

Response: The glycans in this figure are pure commercial substrates and the activity was monitored using TLC analysis. This is clarified in the legend text. The clarified cell lysates should not contain activity unless if the enzymes are either intracellular or periplasmic. The cell-attached enzymes are typically anchored to the lipids in the membrane, so they should theoretically not partition to the soluble proteome in the clarified lysate. We agree with the reviewer that activity in the cell debris would be expected, but this was not detected possibly due to the loss of integrity or inhibition in the cell-debris fraction. This, however, does not change the clear activity data from the intact cell fractions providing supporting the cell attachment claim

- The sentence “fucosidase and sialidase was critical for mucin destruction” is not supported by Fig 4A, B, or the data in figure 4. Please remove or elaborate on how fig 4 shows this.

Answer: The inhibition of fucosidase and sialidase activities abolished growth on mucin but not on a minimal medium, which contained a proteinaceous nitrogen source and monosaccharides. These data provide evidence that the fucosidases and sialidases are crucial to initiate growth on mucin, likely granting access to the underlying glycan structures to be further degradation by glycosidases or by glycopeptidases. The exposure of protease sites for glycopeptidases by sialidases, which is now inserted in the discussion and an additional Fig. 5, summarise the model based on our data and available literature. At any rate, the data show that when the decapping enzymes are inhibited, the organism is unable to access mucin. If the bacterium was able to access either any nutrients, growth on mucin should have been observed. Our data are also supported by a reprint showing that mutations in one of the sialidase genes impairs growth. This is also added to the manuscript

REVIEWERS' COMMENTS

Reviewer #1 (Remarks to the Author):

I think the authors have responded well to my previous comments and I appreciate the clarifying experiments on the co-cultures.

I really like the new Figure 5, but please correct the spelling of glycopeptidases (glycopetidases) on the bottom of the figure.

Also, I would suggest updating Firmicutes to Bacillota, possibly by "Bacillota (formerly Firmicutes)", since this new name will likely be more used in future studies referring to this one.